# Advancing peristalsis deciphering in mouse small intestine by multi-parameter tracking

Anusree Sasidharan[1,5], Breman Anil Peethambar[1,5], Keerthi Santhosh Kumar[2], Ashok V. Kumar [3], Arun Hiregange[2], Neil Fawkes [2], James F. Collins[4], Astrid Grosche[1] & Sadasivan Vidyasagar [1✉]

Assessing gastrointestinal motility lacks simultaneous evaluation of intraluminal pressure (ILP), circular muscle (CM) and longitudinal muscle (LM) contraction, and lumen emptying. In this study, a sophisticated machine was developed that synchronized real-time recordings to quantify the intricate interplay between CM and LM contractions, and their timings for volume changes using high-resolution cameras with machine learning capability, the ILP using pressure transducers and droplet discharge (DD) using droplet counters. Results revealed four distinct phases, $B_{Phase}$, $N_{Phase}$, $D_{Phase}$, and $A_{Phase}$, distinguished by pressure wave amplitudes. Fluid filling impacted LM strength and contraction frequency initially, followed by CM contraction affecting ILP, volume, and the extent of anterograde, retrograde, and segmental contractions during these phases that result in short or long duration DD. This comprehensive analysis sheds light on peristalsis mechanisms, understand their sequence and how one parameter influenced the other, offering insights for managing peristalsis by regulating smooth muscle contractions.

[1] Department of Radiation Oncology, University of Florida, Gainesville, FL, USA. [2] Entrinsic Bioscience, Norwood, MA, USA. [3] Department of Mechanical and Aerospace Engineering, University of Florida, Gainesville, FL, USA. [4] Food Science and Human Nutrition Department, University of Florida, Gainesville, FL, USA. [5] These authors contributed equally: Anusree Sasidharan, Breman Anil Peethambar. ✉email: vidyasagar@ufl.edu

The complex and intricate motility patterns of the gastro-intestinal (GI) tract facilitate the proximal to distal movement of ingested food[1,2]. Peristalsis and segmentation are two distinct forms of motility that facilitate in thorough mixing, digesting, absorbing, and final excreting of foodstuffs[3–5]. Peristalsis involves involuntary contractions of both circular muscle (CM) and longitudinal muscle (LM) layers, propelling chyme from oral to aboral direction. In contrast, segmental contractions driven by CM facilitate food mixing[3,5,6]. The coordinated actions between CM and LM is key for proper food propulsion, and any disruption can lead to GI motility disorders[7].

GI motility disorders refer to conditions that affect the normal movement of food or digesta through the digestive tract and can affect any part of the GI tract from the esophagus to the anus. The underlying cause may often be considered as unknown or multifactorial, involving a combination of genetic, environmental and lifestyle factors. Although the underlying mechanisms and complex interplay of factors involved in GI motility have been explained in previous studies[7–10], the management of GI motility disorders is still complicated by an incomplete understanding of the precise mechanisms coordinating GI smooth muscle activity, which are necessary for proper bowel emptying.

Our collective understanding of mechanisms underlying gut motility has improved exponentially from new, innovative ex vivo and in vivo approaches to study GI motility, including: (1) high resolution manometry, which records pressure changes at various locations in the intestine; (2) spatiotemporal mapping, which allows analysis of CM and LM contractions; (3) fluoroscopy and MRI, which provide real-time imaging of gut contractions; and (4) in vivo quantification of electrolyte and fluid flux using a number of experimental techniques[9–14]. Due to technical limitations, however, all these parameters have not heretofore been studied simultaneously. Doing so would be a major scientific advance since it would allow comprehensive assessment of how CM and LM work together to appropriately modulate intraluminal pressure changes, anterograde, segmental and retrograde contractions to result in droplet discharge (DD).

The research described in this communication was designed to conceptualize and develop a sophisticated machine that integrates multiple components and subsystems to allow comprehensive analysis of gut motility. One major goal was to be able to accurately quantify the intricate interplay between CM and LM movements, as well as the precise timing of their contractions for resultant DD. In achieving this goal, we hoped to improve the understanding of how numerous physiological factors such as intraluminal pressure (ILP), peristalsis, segmental and longitudinal contractions, absorptive or secretory state of the intestine, and DD at the aboral end work in tandem to maintain GI motility. This knowledge could ultimately lead to the development of novel therapeutic approaches for GI motility disorders.

## Results

### Droplet discharge primarily occurred in $D_{phase}$ or $A_{phase}$.
In the "Short-D group" [$S_D$; short duration droplet discharge occurring in the "During Phase" (phase during high amplitude pressure waves; $D_{Phase}$)], the DDs occurred during the $D_{Phase}$ lasting 0.16 to 0.63 min (range) with an experimental DD duration divided by standard DD mean (D/Ds) of $0.85 \pm 0.05$. The "Long-A group" [$L_A$; long duration droplet discharge occurring in "After Phase" (phase after high amplitude pressure waves; $A_{Phase}$)] exhibited significantly longer duration (range: 0.64–0.98 min), with a D/Ds of $1.09 \pm 0.05$ (range: 0.98–1.8). $L_A$ group exhibited longer durations compared to $S_D$ group. Similarly, the "Long-D group" ($L_D$; long duration droplet discharge occurring in $D_{Phase}$) group had all phases longer than $S_D$ (range: 0.68–0.81 min) and a

significant D/Ds increase ($1.24 \pm 0.04$, $p < 0.001$). $L_D$'s $D_{Phase}$ was longer than $L_A$ ($0.27 \pm 0.02$ min vs. $0.33 \pm 0.05$ min; $p = NS$). These three discharge groups represent distinct lumen filling stages.

### Pressure contraction strength changes during four phases in $S_D$, $L_A$ and $L_D$ groups.
Comparing phases across discharge groups, $L_A$ showed a higher pressure contraction strength ($Ps$; amplitude of pressure contraction) in the "Before Phase" ($B_{Phase}$; phase before the high amplitude pressure waves) compared to $S_D$ and $L_D$ with significant difference in $S_D$ ($0.12 \pm 0.01$ cmH$_2$O vs. $0.22 \pm 0.02$ cmH$_2$O, $p < 0.001$; Table 1). $Ps$ change between phases was smaller in $L_A$ compared to $S_D$ and $L_D$. Results suggested substantial $Ps$ increase coincided with both $D_{Phase}$ and $A_{Phase}$ fluid discharges. $L_A$ group's fluid discharge featured a relatively modest $Ps$ increase. Nonetheless, whether $Ps$ of $D_{Phase}$ alone accounted for DD duration remained unclear.

### Gross pressure contraction changes during four phases in $S_D$, $L_A$ and $L_D$ groups.
Gross pressure contraction's ($Pg$; mean pressure changes between two adjacent phases) influence on DD was examined by analyzing its changes between phases (Table 1). In $S_D$ group, $Pg$ significantly increased from $B_{Phase}$ to "Near Phase" ($N_{Phase}$; phase near high amplitude pressure waves, after $B_{Phase}$), and from $N_{Phase}$ to $D_{Phase}$ followed by a decrease from $D_{Phase}$ to $A_{Phase}$. Maximum increase occurred from $N_{Phase}$ to $D_{Phase}$ ($0.05 \pm 0.01$ cmH$_2$O vs. $0.27 \pm 0.06$ cmH$_2$O; $p < 0.02$, $n = 18$). The decrease in $Pg$ from $D_{Phase}$ to $A_{Phase}$ was significant compared to $B_{Phase}$ to $N_{Phase}$ and $N_{Phase}$ to $D_{Phase}$ ($-0.28 \pm 0.06$ cmH$_2$O; $p < 0.001$, $n = 18$). This study revealed an association between DD in $D_{Phase}$ and high $Pg$ from $N_{Phase}$ to $D_{Phase}$.

Comparing $B_{Phase}$ to $N_{Phase}$ and $D_{Phase}$ to $A_{Phase}$, $S_D$ ($p < 0.04$) and $L_D$ ($p < 0.002$) groups had significantly higher $Pg$ than $L_A$. $Pg$ significantly decreased in all three DD groups in $D_{Phase}$ to $A_{Phase}$. Despite significant pressure changes between phases, $L_A$'s $Pg$ remained small, potentially contributing to longer durations between discharges. However, this observation did not fully explain $L_D$ discharges at relatively higher $Pg$ than $L_A$ group. Consequently, $Pg$ increase could only account for $D_{Phase}$ discharges and not discharge duration.

### Gross longitudinal movement changes during four phases in $S_D$, $L_A$ and $L_D$ groups.
The magnitude by which LM moved from the oral end during $B_{Phase}$ to $D_{Phase}$ ($0.20 + 0.14$ mm) was ~ equal to the distance moved from $D_{Phase}$ to $A_{Phase}$ ($-0.33 \pm 0.06$ mm) toward the aboral end.

Although gross longitudinal movement changes ($Lg$; mean longitudinal movement between two adjacent phases) increased from $N_{Phase}$ to $D_{Phase}$ and rapidly decreased from $D_{Phase}$ to $A_{Phase}$, the $L_D$ showed no significant differences between $B_{Phase}$ to $N_{Phase}$, $N_{Phase}$ to $D_{Phase}$ and $D_{Phase}$ to $A_{Phase}$ (Table 1).

In $L_A$ group, $Lg$ was lower in all three phases, with significant reduction in $B_{Phase}$ to $N_{Phase}$ and $D_{Phase}$ to $A_{Phase}$ compared to $S_D$ group, suggesting reduced LM contraction. Similarly, $L_D$ group exhibited lower $Lg$ in $B_{Phase}$ to $N_{Phase}$ and $D_{Phase}$ to $A_{Phase}$ compared to $S_D$ group (Table 2). However, $L_D$ group showed no significant difference between phases and notably less aboral movement compared to $S_D$ group ($-0.07 \pm 0.06$ mm vs. $-0.33 \pm 0.06$ mm; $p < 0.02$). $L_D$ group also showed minor aboral movement in $B_{Phase}$ to $N_{Phase}$ compared to different phases ($p = NS$). Reduced LM movement toward both oral and /or aboral ends in $L_A$ and $L_D$ groups might contribute to the longer duration of these DDs.

**Table 1 Gross movement and amplitude strength analysis of ILP (intraluminal pressure; referring to $Pg$ and $Ps$), longitudinal muscle movement (referring to $Lg$ and $Ls$), edge width 1 diameter ($EW_1i$; referring to $EW_1g$ and $EW_1s$), edge width 4 diameter (referring to $EW_4g$ and $EW_4s$), and volume (referring to $Vg$ and $Vs$) between various phases of the different droplet discharge groups.**

| | | Gross movement | | | | Amplitude strength | | | |
|---|---|---|---|---|---|---|---|---|---|
| | | $B$ to $N_{Phase}$ | $N$ to $D_{Phase}$ | $D$ to $A_{Phase}$ | | $B_{Phase}$ | $N_{Phase}$ | $D_{Phase}$ | $A_{Phase}$ |
| $Pg$ (cmH$_2$O) | S$_D$ | $0.05 \pm 0.01^a$ | $0.27 \pm 0.06^b$ | $-0.28 \pm 0.06^c$ | $Ps$ (cmH$_2$O) | $0.12 \pm 0.01^a$ | $0.28 \pm 0.04^a$ | $0.72 \pm 0.15^b$ | $0.19 \pm 0.04^{a,c}$ |
| | L$_A$ | $-0.01 \pm 0.02^{*,a}$ | $0.16 \pm 0.05^b$ | $-0.08 \pm 0.06^{*,a,c}$ | | $0.20 \pm 0.02^{*,a}$ | $0.26 \pm 0.03^a$ | $0.47 \pm 0.04^b$ | $0.25 \pm 0.03^{a,c}$ |
| | L$_D$ | $0.11 \pm 0.02^{*,\#,a}$ | $0.19 \pm 0.03^{a,b}$ | $-0.27 \pm 0.04^{\#,c}$ | | $0.16 \pm 0.03^a$ | $0.31 \pm 0.04^a$ | $0.93 \pm 0.25^b$ | $0.27 \pm 0.08^{a,c}$ |
| $Lg$ (mm) | S$_D$ | $0.20 \pm 0.04^a$ | $0.14 \pm 0.05^a$ | $-0.33 \pm 0.06^b$ | $Ls$ (mm) | $0.51 \pm 0.07^a$ | $0.88 \pm 0.12^{ab}$ | $1.10 \pm 0.14^b$ | $0.60 \pm 0.08^{a,c}$ |
| | L$_A$ | $0.03 \pm 0.03^{*,a}$ | $0.06 \pm 0.02^a$ | $-0.09 \pm 0.02^{*,b}$ | | $0.50 \pm 0.1$ | $0.54 \pm 0.07^*$ | $0.63 \pm 0.07^{*,}$ | $0.53 \pm 0.06$ |
| | L$_D$ | $-0.01 \pm 0.02^*$ | $0.12 \pm 0.12$ | $-0.07 \pm 0.06^*$ | | $0.82 \pm 0.20$ | $0.94 \pm 0.18^\#$ | $1.27 \pm 0.25^{*,\#}$ | $0.99 \pm 0.15^{*,\#}$ |
| $EW_1g$ (mm) | S$_D$ | $-0.002 \pm 0.004^a$ | $-0.13 \pm 0.02^b$ | $0.13 \pm 0.03^c$ | $EW_1s$ (mm) | $0.22 \pm 0.02^a$ | $0.26 \pm 0.02^a$ | $0.14 \pm 0.01^b$ | $0.23 \pm 0.02^a$ |
| | L$_A$ | $-0.01 \pm 0.01$ | $-0.02 \pm 0.02^*$ | $0.02 \pm 0.02^*$ | | $0.19 \pm 0.04$ | $0.20 \pm 0.04^\#$ | $0.18 \pm 0.03$ | $0.19 \pm 0.03$ |
| | L$_D$ | $-0.02 \pm 0.04$ | $-0.02 \pm 0.04^*$ | $0.03 \pm 0.05$ | | $0.29 \pm 0.09$ | $0.33 \pm 0.06$ | $0.33 \pm 0.07^{*,\#}$ | $0.31 \pm 0.09$ |
| $EW_4g$ (mm) | S$_D$ | $0.02 \pm 0.02^a$ | $-0.06 \pm 0.02^b$ | $0.01 \pm 0.02^a$ | $EW_4s$ (mm) | $0.16 \pm 0.02^a$ | $0.18 \pm 0.03^a$ | $0.09 \pm 0.02^{a,b}$ | $0.20 \pm 0.03^a$ |
| | L$_A$ | $0.02 \pm 0.01$ | $-0.02 \pm 0.02$ | $0.02 \pm 0.02$ | | $0.18 \pm 0.04$ | $0.19 \pm 0.05$ | $0.12 \pm 0.04$ | $0.2 \pm 0.05$ |
| | L$_D$ | $0.02 \pm 0.01^a$ | $-0.07 \pm 0.04^b$ | $0.02 \pm 0.01^b$ | | $0.23 \pm 0.05$ | $0.25 \pm 0.06,$ | $0.19 \pm 0.05$ | $0.23 \pm 0.05$ |
| $Vg$ (mm$^3$) | S$_D$ | $-0.03 \pm 0.05^a$ | $-0.65 \pm 0.13^a$ | $0.59 \pm 0.15^b$ | $Vs$ (mm$^3$) | $9.28 \pm 1.57$ | $12.59 \pm 2.49$ | $8.56 \pm 1.52$ | $9.5 \pm 1.91$ |
| | L$_A$ | $-0.01 \pm 0.08^a$ | $0.07 \pm 0.08^{*,a}$ | $-0.1 \pm 0.09^{*,b}$ | | $6.41 \pm 1.26$ | $6.94 \pm 1.48^*$ | $7.39 \pm 1.68$ | $7.51 \pm 1.56$ |
| | L$_D$ | $0.27 \pm 0.17$ | $-0.12 \pm 0.17^*$ | $0.13 \pm 0.08^{*,\#}$ | | $11.74 \pm 3.7$ | $17.47 \pm 5.49^\#$ | $15.42 \pm 4.64$ | $14.73 \pm 5.49$ |

Significant differences between S$_D$, L$_A$ and L$_D$ in corresponding phases were calculated using one-way ANOVA followed by Bonferroni for individual comparison: * represents significant differences to S$_D$, # represents significant differences to L$_A$; significant differences between $B_{Phase}$, $N_{Phase}$, $D_{Phase}$ and $A_{Phase}$ in corresponding variables: different letters (a, b, c) indicate significant differences between phases ($p < 0.05$). The number of data for each comparison are accessible in Supplementary Data[51].

**Longitudinal contraction strength changes during four phases in S$_D$, L$_A$ and L$_D$ groups.** In the S$_D$ group, the longitudinal contraction strength ($Ls$; amplitude of longitudinal contractions) progressively increased from $B_{Phase}$ to $D_{Phase}$, then declined in $A_{Phase}$. In S$_D$ group, $Ls$ in $D_{Phase}$ was significantly higher compared to both $B_{Phase}$ and $A_{Phase}$. Conversely, L$_A$ group showed no significant differences in Ls between phases. In L$_A$ group, $Ls$ in $D_{Phase}$ was significantly lower compared to S$_D$ group. L$_A$ group displayed small non-significant differences in $Ls$ between phases. Conversely, in L$_D$ group, $Ls$ in $D_{Phase}$ was significantly higher compared to S$_D$ group. Although L$_D$ generally had higher $Ls$, it lacked significant phase distinctions, potentially contributing to increased DD duration. Furthermore in L$_D$ group, $Ls$ was significantly higher compared to L$_A$ group in all phases except $B_{Phase}$ (Table 1).

**Gross volume and volume strength changes during four phases in S$_D$, L$_A$ and L$_D$ groups.** In the S$_D$ and L$_A$ groups, intestinal gross volume ($Vg$; mean volume changes between two adjacent phases) decreased marginally from $B_{Phase}$ to $N_{Phase}$ and $N_{Phase}$ to $D_{Phase}$. S$_D$ group showed a significant volume drop from $N_{Phase}$ to $D_{Phase}$, unlike L$_A$ group. S$_D$ group showed significantly increased volume in $D_{Phase}$ to $A_{Phase}$ (Table 1). Conversely, L$_A$ group displayed minimal changes, with an increase from $N_{Phase}$ to $D_{Phase}$ followed by a slight decrease in $D_{Phase}$ to $A_{Phase}$. This suggests delayed fluid filling and emptying in L$_A$ compared to S$_D$ group. L$_D$ exhibited non-significant volume increments between phases unlike S$_D$ group. This explains the delayed lumen filling and prolonged duration of DD in L$_D$ group. Taking all phases together, S$_D$ ($9.99 \pm 0.96$ mm$^3$; $p < 0.001$, $n = 72$) and L$_D$ ($14.84 \pm 2.35$ mm$^3$; $p < 0.006$, $n = 32$) exhibited higher volume strength ($Vs$; amplitude of volume contraction) compared to L$_A$ ($7.06 \pm 0.73$ mm$^3$, $n = 76$). These studies showed that irrespective of contraction phase, L$_A$ consistently displayed lower $Vs$. Phase-related $Vs$ changes for S$_D$, L$_A$ and L$_D$ showed no significant difference (Table 1).

**Gross movement of edge width tracker 1 and 4 changes during four phases in S$_D$, L$_A$ and L$_D$ groups.** Gross movement of edge width tracker 1 ($EW_1g$; mean diameter changes between adjacent phases in edge width tracker 1; $EW_1$) exhibited marginal decrease in $B_{Phase}$ to $N_{Phase}$ followed by a significant decrease in $N_{Phase}$ and $D_{Phase}$ in S$_D$, signifying proximal CM contraction with DD. Conversely, L$_A$ and L$_D$ had minimal $EW_1g$ changes, suggesting DD without significant CM contractions, explaining delayed emptying.

Gross movement of edge width tracker 4 ($EW_4g$; mean diameter changes between adjacent phases in edge width tracker 4; $EW_4$) increased slightly from $B_{Phase}$ to $N_{Phase}$ and, decreased significantly from $N_{Phase}$ to $D_{Phase}$ in S$_D$, implying open lumen becoming closed. A small non-significant increase followed from $D_{Phase}$ to $A_{Phase}$. L$_A$ showed smaller insignificant diameter changes. L$_D$ displayed a small but significant diameter decrease from $B_{Phase}$ to $N_{Phase}$ compared to $N_{Phase}$ to $D_{Phase}$, suggesting closed to open lumen transition. The decreased $B_{Phase}$ to $N_{Phase}$ diameter could explain L$_D$ duration.

Comparing $EW_1g$ and $EW_4g$ contractions, a significant diameter decreases in $EW_1g$ during $N_{Phase}$ to $D_{Phase}$ suggested proximal contraction with distal CM relaxation. In S$_D$, lumen filling primarily occurred in $D_{Phase}$ to $A_{Phase}$, where $EW_1g$ diameter increased compared to $EW_4g$ (Table 2). Similar changes were absent in the L$_A$ and L$_D$ groups, although $EW_1g$ showed greater diameter decrease than $EW_4g$.

**Table 2 Frequency comparison of intraluminal pressure (ILP), longitudinal muscle (LM) movement, edge width 1 diameter ($EW_1$), edge width 4 diameter ($EW_4$) and volume within the various phases of the different droplet discharge groups.**

| Frequency (Hz) | | $B_{Phase}$ | $N_{Phase}$ | $D_{Phase}$ | $A_{Phase}$ |
|---|---|---|---|---|---|
| ILP | $S_D$ | $0.74 \pm 0.01$ | $0.73 \pm 0.01$ | $0.72 \pm 0.01$ | $0.76 \pm 0.03$ |
| | $L_A$ | $0.69 \pm 0.01^*$ | $0.66 \pm 0.02^*$ | $0.66 \pm 0.01^*$ | $0.67 \pm 0.01^*$ |
| | $L_D$ | $0.73 \pm 0.02$ | $0.71 \pm 0.02^{\#}$ | $0.70 \pm 0.01^{\#}$ | $0.70 \pm 0.02$ |
| LM | $S_D$ | $0.74 \pm 0.01$ | $0.73 \pm 0.01$ | $0.72 \pm 0.01$ | $0.72 \pm 0.01$ |
| | $L_A$ | $0.69 \pm 0.01^*$ | $0.65 \pm 0.01^*$ | $0.63 \pm 0.02^{*,\#}$ | $0.62 \pm 0.02^{*,\#}$ |
| | $L_D$ | $0.74 \pm 0.01^{\#}$ | $0.72 \pm 0.02^{\#}$ | $0.72 \pm 0.01^{\#}$ | $0.71 \pm 0.01^{\#}$ |
| $EW_1$ | $S_D$ | $0.74 \pm 0.01$ | $0.73 \pm 0.01$ | $0.73 \pm 0.01$ | $0.73 \pm 0.01$ |
| | $L_A$ | $0.69 \pm 0.01^*$ | $0.69 \pm 0.01^*$ | $0.67 \pm 0.01^*$ | $0.67 \pm 0.01^*$ |
| | $L_D$ | $0.73 \pm 0.01$ | $0.73 \pm 0.02$ | $0.73 \pm 0.02^{\#}$ | $0.72 \pm 0.01^{\#}$ |
| $EW_4$ | $S_D$ | $0.70 \pm 0.02$ | $0.72 \pm 0.02$ | $0.70 \pm 0.01$ | $0.72 \pm 0.01$ |
| | $L_A$ | $0.68 \pm 0.02$ | $0.65 \pm 0.02^*$ | $0.65 \pm 0.01^*$ | $0.67 \pm 0.01^*$ |
| | $L_D$ | $0.80 \pm 0.11$ | $0.68 \pm 0.02$ | $0.70 \pm 0.01^{\#}$ | $0.71 \pm 0.02$ |
| Volume | $S_D$ | $0.74 \pm 0.01$ | $0.75 \pm 0.01$ | $0.73 \pm 0.01$ | $0.73 \pm 0.01$ |
| | $L_A$ | $0.68 \pm 0.01^*$ | $0.67 \pm 0.01^*$ | $0.65 \pm 0.01^*$ | $0.65 \pm 0.01^*$ |
| | $L_D$ | $0.72 \pm 0.02$ | $0.72 \pm 0.02$ | $0.72 \pm 0.01^{\#}$ | $0.70 \pm 0.01^{\#}$ |

Significant differences between $S_D$, $L_A$ and $L_D$ in corresponding phases were calculated using one-way ANOVA followed by Bonferroni for individual comparison: * represents significant differences to $S_D$, # represents significant differences to $L_A$ ($p < 0.05$). The number of data for each comparison are accessible in Supplementary Data[51].

**Changes in proximal and distal edge width contraction strength during four phases in $S_D$, $L_A$ and $L_D$ groups.** In $S_D$, proximal edge width contraction strength ($EW_1s$; amplitude of $EW_1$ contraction) significantly decreased in $D_{Phase}$ compared to $B_{Phase}$ (Table 1). This suggests maximum CM contractions during $N_{Phase}$ to $D_{Phase}$ transition when $EW_1s$ was lowest. $L_A$ and $L_D$ showed stable $EW_1s$ with no significant phase differences. $L_D$ generally had higher $EW_1s$ compared to $L_A$ with significance in $D_{Phase}$. DD in $D_{Phase}$ followed $EW_1s$ and $EW_1g$ decrease, suggesting a role for proximal CM contraction. Unlike $EW_1s$, mean diameter for all phases was lower in $EW_4s$ for $S_D$ ($0.21 \pm 0.01$ mm vs. $0.16 \pm 0.01$ mm; $p < 0.002$, $n = 72$) and $L_D$ ($0.32 \pm 0.04$ mm vs. $0.23 \pm 0.02$ mm; $p < 0.05$, $n = 32$). In $L_A$, there was no significant difference between $EW_1s$ and $EW_4s$. In $L_A$ and $L_D$, there was no significant phase differences. In $S_D$, distal edge width contraction strength ($EW_4s$; amplitude of $EW_4$) increased from $B_{Phase}$ to $N_{Phase}$, decreased in $D_{Phase}$, and significantly increase in $A_{Phase}$ ($p < 0.03$, $n = 18$) with maximum $EW_1s$ and $EW_4s$ decrease occurring in $D_{Phase}$.

**Frequency measurements during different contraction phases and droplet discharge intraluminal pressure wave frequency.** Comparing ILP frequency across phases of DD groups showed no significant differences. However, distinct pattern emerged when comparing $S_D$, $L_A$ and $L_D$ groups (Table 2). Frequency decrease was significant between $S_D$ and $L_A$ in $B_{Phase}$ ($0.74 \pm 0.01$ Hz vs. $0.69 \pm 0.01$ Hz; $p < 0.02$), $N_{Phase}$ ($0.73 \pm 0.01$ Hz vs. $0.66 \pm 0.02$ Hz; $p < 0.001$), $D_{Phase}$ ($0.72 \pm 0.01$ Hz vs. $0.66 \pm 0.01$ Hz; $p < 0.003$) and $A_{Phase}$ ($0.76 \pm 0.03$ Hz vs. $0.67 \pm 0.01$ Hz; $p < 0.001$). This suggests that frequency decreases as contraction decreases, akin to $L_A$. Intriguingly, $L_D$ exhibited significantly higher frequencies compared to $L_A$ in $N_{Phase}$ ($0.71 \pm 0.02$ Hz vs. $0.66 \pm 0.02$ Hz; $p < 0.05$) and $D_{Phase}$ ($0.70 \pm 0.01$ Hz vs. $0.66 \pm 0.01$ Hz; $p < 0.04$), and were not significantly different from $S_D$ group.

**Longitudinal tracker frequency analysis.** Comparing frequencies of LM contractions within $S_D$, $L_A$ and $L_D$ phases showed no significant differences (Table 2), suggesting consistent pacemaker activity across discharge group phases. However, overall frequency and mean amplitude analysis using fast Fourier transform (FFT) revealed higher values in $S_D$ than $L_A$ (Fig. 1a, b). Frequencies between $S_D$ and $L_A$ phases displayed significantly lower values in $L_A$ group across all phases.

Similarly, $L_A$ exhibited significantly lower frequency in all phases compared to the $L_D$ group (Table 2). There was no significant difference in longitudinal frequencies between $S_D$ and $L_D$ phases. In summary, these findings suggest that when the intestinal lumen is mostly empty ($L_A$), the contraction frequencies are lower across all four phases. Conversely, in $S_D$ and $L_D$ the frequencies are higher across all phases.

**Proximal edge width tracker frequency analysis.** Comparing frequencies within $S_D$, $L_A$ and $L_D$ phases of $EW_1$ showed no significant difference. However, significant distinctions emerged between discharge groups. $S_D$ exhibited higher frequency compared to $L_A$ across all phases (Table 2). Comparing $L_A$ and $L_D$, $L_A$ displayed significantly lower frequencies in $D_{Phase}$ and $A_{Phase}$. While $B_{Phase}$ and $N_{Phase}$ in $L_A$ displayed lower frequencies compared to $L_D$ but was not significant. These studies showed that lower frequencies in the $L_A$ group aligned with minimal diameter changes between phases (Table 2), suggesting CM contraction frequency has a potential role in fluid discharge dynamics.

**Distal edge width tracker frequency analysis.** Frequency analyses of $S_D$, $L_A$ and $L_D$ revealed no significant difference between phases (Table 2). Comparing $S_D$ and $L_A$, $L_A$ showed significantly lower frequencies in $N_{Phase}$, $D_{Phase}$ and $A_{Phase}$. Similarly, $L_A$ exhibited lower frequencies compared to $L_D$ in all phases, with a significant difference in $D_{Phase}$. However, there were no significant differences in frequency between $S_D$ and $L_D$ in all four phases.

When comparing overall $EW_1$ and $EW_4$ frequencies, a significantly higher frequency was observed in $EW_1$ ($0.71 \pm 0.01$ Hz vs. $0.67 \pm 0.00$ Hz; $p < 0.001$, $n = 180$; Fig. 1c). This frequency difference was consistent across various phases of the contractions for $S_D$, $L_A$ and $L_D$ but decreased significantly in $L_A$ which prolonged the time for discharge, particularly in $EW_1$, highlighting the significance of proximal contractions in fluid propulsion dynamics.

Additionally, the frequency pattern indicated stronger proximal contractions, favouring anterograde propulsion. Reduced $EW_4$ frequency relative to $EW_1$ could be influenced by the steady oral fluid flow from the peristaltic pump, contrasting with variable fluid flow in aboral end because of pooling and emptying. This differential exposure to fluid flow could affect contraction frequency and strength.

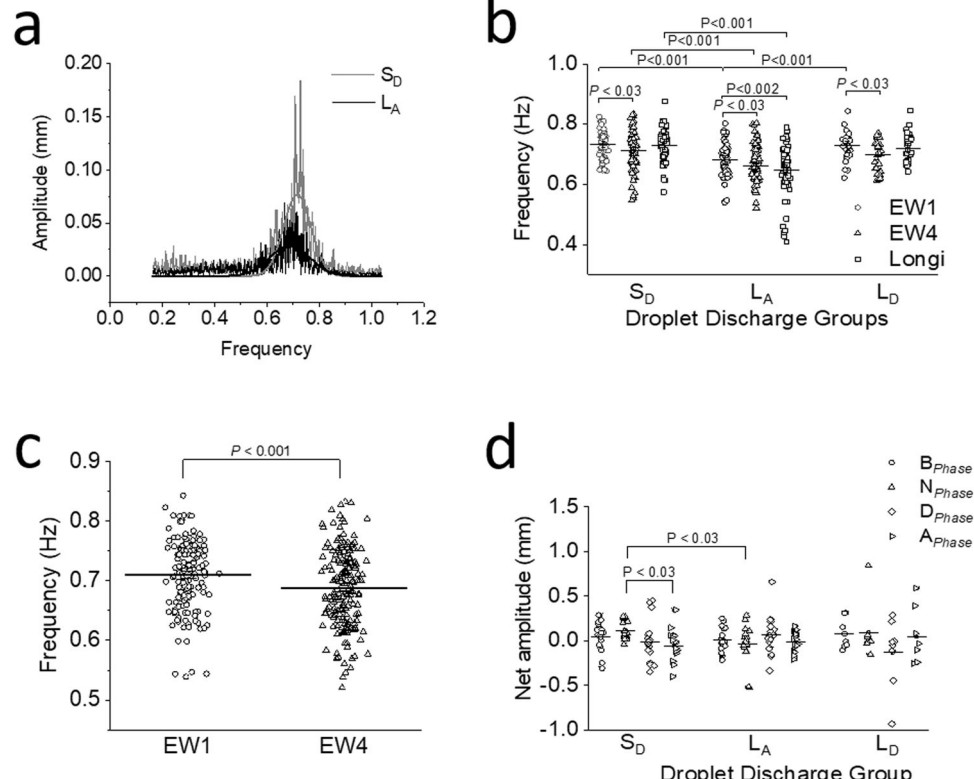

**Fig. 1 Pattern of frequency, contraction types and net amplitude within droplet charge groups. a** Representative graph showing fast Fourier transform (FFT) of longitudinal trace. The sinusoidal oscillations of longitudinal contractions for a period were divided into distinct frequency components with their respective amplitude. A high-pass filter with a cut off frequency of 0.2 Hz and an amplitude threshold of 0.02 mm was used. Non-linear exponential curve fit using second order polynomial exponent (Exp3P2) was used to trace along the FFT. The graph shows a shift in the frequency to the left for $L_A$ (black tracings) compared to $S_D$ (gray tracings), suggesting a decrease in frequency in $L_A$. The decrease in frequency was associated with a simultaneous decrease in the amplitude of the contraction. **b** Comparison of $EW_1$, $EW_4$ and longitudinal frequency within various discharge groups. $EW_1$ and $EW_4$ showed a significant difference in frequencies (Longi) in $S_D$ ($p < 0.03$, $n = 18$), $L_A$ ($p < 0.03$, $n = 19$) and $L_D$ ($p < 0.03$, $n = 8$) discharge groups. Comparing $EW_1$, $EW_4$ and Longi of $S_D$ with that of $L_A$ showed a significant difference. $L_A$ showed significant difference between $EW_1$ and Longi. In $L_D$ discharge groups, there was no statistical significance between frequencies for $EW$ and Longi. Significant differences were calculated using Kruskal–Wallis followed by Mann–Whitney test for individual comparison. **c** Overall frequency comparison between proximal ($EW_1$) and distal edge trackers ($EW_4$) irrespective of the discharge group. Proximal edge tracker have a higher frequency compared to the distal tracker ($p < 0.001$, $n = 48$). Significant differences were calculated using one-way ANOVA. **d** Comparison of net amplitude between $EW_1$ and $EW_4$ for the three discharge groups. Net amplitude was anterograde in $B_{Phase}$ and $N_{Phase}$ in $S_D$, followed by retrograde net movement in $D_{Phase}$ and $A_{Phase}$ with a significant difference between $N_{Phase}$ and $A_{Phase}$ ($p < 0.03$, $n = 18$). In $L_A$, the net amplitude was generally retrograde, with anterograde movement only in $D_{Phase}$ ($n = 19$). In $L_D$, similar to $S_D$, net amplitude was anterograde in $B_{Phase}$ and $N_{Phase}$, retrograde movements were seen only in $D_{Phase}$ followed by a small anterograde movement in $A_{Phase}$ ($n = 8$). In $L_A$ and $L_D$, there are no significant differences between the phases. Significant differences were calculated using Kruskal–Wallis followed by Mann–Whitney test for individual comparison.

Comparing LM and $EW_1$ frequencies, $S_D$ and $L_D$ showed no significant difference across phases. In $L_A$, LM frequency significantly decreased compared to $EW_1$ in $N_{phase}$ ($0.65 \pm 0.01$ Hz vs. $0.69 \pm 0.01$ Hz; $p < 0.04$, $n = 19$) and $A_{phase}$ ($0.62 \pm 0.02$ Hz vs. $0.67 \pm 0.01$ Hz; $p < 0.04$, $n = 19$).

**Anterograde, retrograde, and segmental contractions**
*$S_D$ group.* The "net amplitude" increased from $B_{Phase}$ to $N_{Phase}$ ($0.05 \pm 0.04$ mm vs. $0.12 \pm 0.02$ mm), and declined during $D_{Phase}$ ($-0.01 \pm 0.05$ mm) and $A_{Phase}$ ($-0.05 \pm 0.04$ mm) with a significant difference between $N_{Phase}$ and $A_{Phase}$ ($p < 0.03$) (Fig. 1d). This indicates that strong proximal contractions facilitate DD, with anterograde contractions during $B_{Phase}$ to $N_{Phase}$, while retrograde contractions appeared in $D_{Phase}$ and $A_{Phase}$.

*$L_A$ group.* "Net amplitude" was relatively lower compared to $S_D$. Retrograde contractions were noted in $N_{Phase}$ ($-0.03 \pm 0.05$ mm) and $A_{Phase}$ ($-0.01 \pm 0.02$ mm). Low amplitude anterograde

movements occurred in $B_{Phase}$ ($0.01 \pm 0.03$ mm) and $D_{Phase}$ ($0.07 \pm 0.05$ mm), without significant differences between the four phases (Fig. 1d).

*$L_D$ group.* $L_D$ had larger amplitude contractions than $L_A$. $L_D$ exhibited anterograde contractions in all phases except $D_{Phase}$, marked by retrograde contractions (Fig. 1d), without any significant differences between phases.

*Comparing $S_D$, $L_A$, and $L_D$.* $L_A$ had lower contraction amplitude with a significant difference only in $N_{Phase}$ ($0.12 \pm 0.02$ mm vs. $-0.03 \pm 0.05$ mm; $p < 0.004$, $n = 18$ and $n = 19$, respectively; Fig. 1d) when compared to $S_D$. No significant differences within phases of $S_D$ and $L_D$ groups were observed.

These findings suggest anterograde contractions precede DD. Variable "net amplitude" between phases in $L_A$ and $L_D$ discharge groups highlights the complexity of intestinal motility. Strong proximal CM and LM contractions and volume decrease in $B_{Phase}$ to $N_{Phase}$ and $N_{Phase}$ to $D_{Phase}$ are responsible for shorter DD

duration, anterograde contraction in $S_D$, while lower amplitude and predominantly retrograde contractions in $L_A$ might contribute to longer DD durations.

## Discussion

Functional imbalance in GI motility is closely linked to numerous human ailments such as achalasia, dyspepsia, gastroparesis, constipation, IBS, chronic intestinal pseudo-obstruction[15]. The current understanding of intestinal peristalsis relies on in vitro and in vivo techniques including manometry and spatiotemporal mapping, but each technique has its own limitations[9,10]. Technical limitations have hindered quantitative analysis of smooth muscle layer movements affecting ILP and DD. In this study, a multi-spectrum image capture system, ILP transducers, Vernier Drop counters and associated proprietary software programs enabled real time acquisition of quantitative data on intestinal motility and DD. Specifically, the technology achieved: (1) Quantification of frequency and amplitude for both, high and low amplitude pressure waves; (2) Continuous monitoring of diameter and volume changes for luminal content accumulation and propulsion; (3) Characterizing intestinal contractions as anterograde, retrograde or segmentation; (4) Establishing "net amplitude" to assess overall contraction strength and direction; (5) Relationship to DD using various motility parameters; and (6) Establishing absorptive or secretory state of the intestine but is not part of this study, as intestinal segments were perfused with ringer solution and exhibited net absorption. Future comparisons between formulations will consider net fluid output, impacting both, motility and, absorption or secretion. Overall, this is an approach that simultaneously uses all these parameters to study the mechanisms of gut motility.

Interstitial cells of Cajal (ICCs) are specialized pacemaker cells that initiate and propagate slow electrical waves regulating smooth muscle contraction frequency in response to force[16–18]. Various ICC types are classified by their GI tract location[19]. They can be interconnected multipolar cells (myenteric region) or bipolar without interconnections within CM and LM layers[17,18,20]. ICC communicate directly via gap junctions and chemical signaling[18]. ICC initiate oscillations, initially synchronized proximally, but increasingly desynchronized aborally[17]. ICC form networks such as the Myenteric plexus situated between the CM and LM layers, coordinating peristalsis and food propulsion, or the Meissner's plexus in the submucosa, coordinating local reflexes in response to various GI stimuli, and modulating peristalsis, vasomotor activity, absorption and secretion[19].

In this study, CM and LM contractions were referenced to four phases of pressure tracing ($B_{Phase}$, $N_{Phase}$, $D_{Phase}$ and $A_{Phase}$) to clarify LM's role in initiating CM contractions, and the resulting volume and pressure changes leading to DD. In $S_D$ group, increased $Lg$ occurs through $B_{Phase}$, $N_{Phase}$ and $D_{Phase}$, followed by CM contraction. This could be explained by CM fiber gathering and associated ICC proximity, facilitating signal transmission and contraction. In dog ileum, LM played a crucial role in transmitting electrical signals to CM upon distention[21,22]. Similarly, findings in human esophagus revealed that LM contractions precede CM contractions, and last longer[23,24]. $EW_1$ began to contract in $B_{Phase}$ to $N_{Phase}$, while $EW_4$ continued diameter increase suggesting relative aboral CM relaxation while LM contraction peaked ahead of CM contraction. Maximum increase in ILP followed $EW$ contraction and decreased volume in $N_{Phase}$ to $D_{Phase}$, rather than LM tracker movement, suggesting ILP changes as a reflection of CM contraction. Similar observations in other studies on intestine and esophagus have shown concurrent contraction and relaxation of CM and LM layers[25,26].

In this study, $EW_4$ was associated with a more relaxed state compared to $EW_1$, and this combined with a relatively higher contraction frequency in the proximal regions, favored more anterograde contraction, irrespective of the DD type. Similar observations were made in animal and human showing that GI tract's distal regions progressively exhibit reduced frequency and contractions[27–31]. Increased frequency and contraction in proximal segments may be necessary for mixing and propelling digested material in the upper GI tract. The frequency of contractions between LM and $EW_1$ did not significantly differ in $S_D$ and $L_D$ but showed a significantly lower frequency in $L_A$ groups. Similar findings were observed in studies done in human ileal tissue[32]. This explains the decreased frequency, and LM and CM contractions when the lumen is mostly empty, as fewer stretch reflexes are initiated to stimulate the myenteric plexus (Fig. 2a).

The "net amplitude" displayed more anterograde contractions and fluid emptying when distal tracker contractions lagged the proximal tracker contraction. Segmental contractions with no fluid emptying occurred in the absence of a lag between the two trackers. This is evident in $S_D$ group, where $B_{Phase}$, $N_{Phase}$ and $D_{Phase}$ were anterograde with a significant difference between $B_{Phase}$ and $A_{Phase}$. Fluid accumulated toward the aboral end, leading to droplet formation. Although, DD happened in $D_{Phase}$ or $A_{Phase}$, the fluid accumulation commenced as early as $B_{Phase}$ continuing through $N_{Phase}$ and the early part of $D_{Phase}$. Eventually, the droplet's weight overcame cohesive forces, causing DD. Thus, anterograde movements occur during fluid emptying phases and shifted to net retrograde movement during fluid filling phases.

Studies suggest that ICCs decide frequency and character of propagative muscle contractions[33]. The proximal edge trackers recorded a diameter reduction from $B_{Phase}$ to the end of $D_{Phase}$, after which the diameter began to increase progressively and peaking in $D_{Phase}$ to $A_{Phase}$. $EW_4$ showed a relatively smaller reduction, indicating less gross contraction. Studies using rat small intestine, showed that proximal CMs were more sensitive to cholinergic drugs and active stress when compared to distal CM contraction[34].

The decrease in $Pg$ and $Ps$ during $A_{Phase}$ correlated with decreased $Lg$, signifying LM movement toward the distal end, accompanied by increased $Vg$ and $Vs$. Following the peak contraction of $EW_1$, LM contracted aborally in $D_{Phase}$ to $A_{Phase}$ and $A_{Phase}$, associated with increased $EW_1$, while $EW_4$ showed only a minor increase. During this phase, the intestinal segment volume increased substantially, indicating fluid filling as LM contracts at the aboral end initiating contraction of $EW_4$. Subsequently, LM moved toward the oral end as the volume continued to fill, and CM relaxed. This suggests that LM contraction is vital for initiation of contraction at both, oral and aboral ends. However, to sustain the CM contraction at either end, LM contraction may not be necessary (Fig. 2b, c).

In $L_D$ group, proximal LM movement was reduced in $N_{Phase}$ to $D_{Phase}$ compared to $S_D$ group but was not significant. $EW_1$ did not show a significant difference between various phases, but was significantly lower compared to $S_D$. These contraction parameters explained the relative retrograde movement observed in $D_{Phase}$ of $L_D$ group, and the relatively longer time for DD. Both, $L_A$ and $L_D$ groups had preceding short duration discharges. $L_A$ group had a longer DD duration compared to $L_D$ group. Consequently, the $L_A$ group had less time to fill the lumen before the next discharge, explaining why $L_D$ group had a higher intestinal volume, significantly in $D_{Phase}$ to $A_{Phase}$. In $S_D$, intestinal volume increased, likely activating the stretch reflex for strong contractions. In $L_A$, delayed filling resulted in less stretch reflex activation, manifesting as lower muscular contractions. $L_D$ was intermediate with

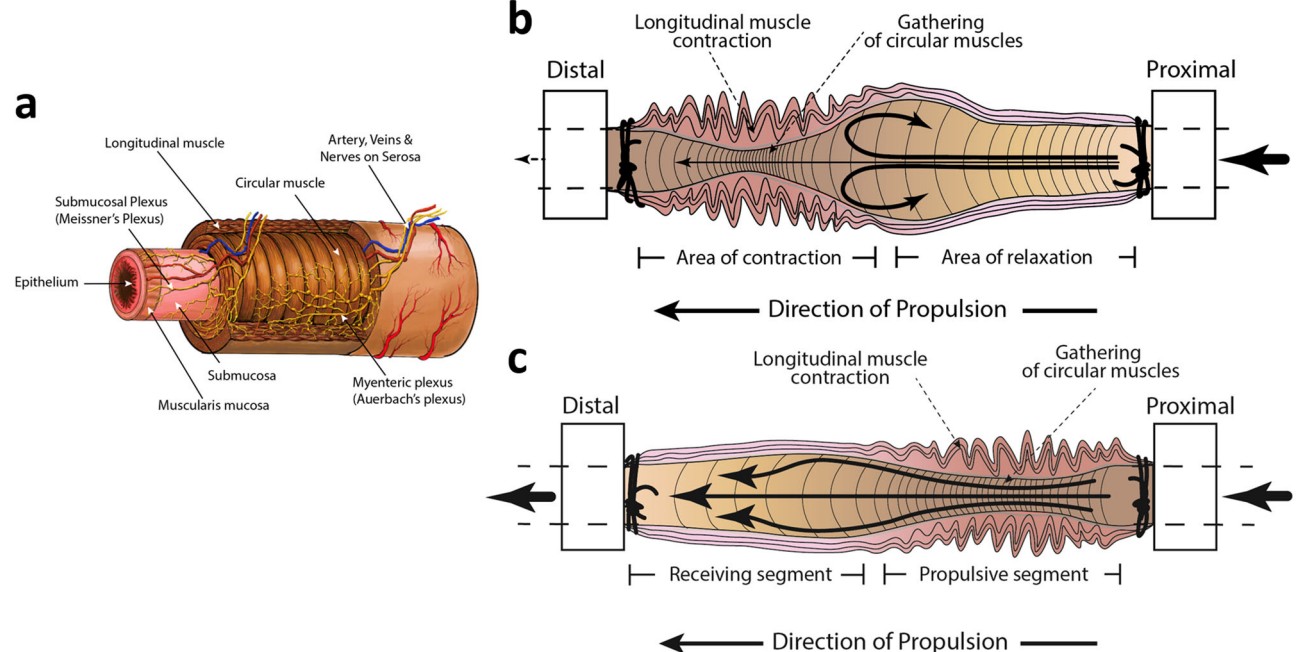

**Fig. 2 Illustration of the small intestine with organization of circular and longitudinal muscles, and the enteric nervous system, and schematic representation of the intestine showing fluid filling and muscle contractions for droplet discharge. a** The illustration shows the entry of artery, veins and nerves from the mesentery through the serosa and the distribution of the myenteric plexus between the longitudinal and circular layers of smooth muscle. The orientation of the myenteric plexus is such that the ganglia (collection of nerve cell bodies) are stretched circumferentially and inter-ganglionic fibers run longitudinally, parallel to the longitudinal muscle fibers to allow transmission of signals and coordination of muscle contractions without exerting excessive traction or tension on the nerve fibers. Meissner's plexus or submucosal plexus is a network of nerve fibers and ganglia located in the loose connective tissue of the submucosal layer of the GI tract. Receptive nerve endings from Meissner's plexus projects into the mucosa as intrinsic primary afferent neurons and transmit the signals to submucosal neurons or Myenteric plexus. **b** Figure showing longitudinal muscle contraction with gathering of circular muscle rings (muscle fibers) toward the distal end that results in contraction with narrowing of the lumen at the aboral end, favouring fluid-filling. Fluid filling leads to ballooning proximal to the contracted region (bold arrows). Intestinal flow is reduced during the filling stage (thin arrow). Arrows within the lumen depicts more retrograde movement. **c** Figure showing longitudinal muscle contraction (parallel lines) with gathering of circular muscle rings (muscle fibers) and narrowing of the lumen at the proximal end that favors droplet discharge (bold arrows). Arrows within the lumen suggest more anterograde movement. **a** was created using Sketchbook® App in iPad (7th generation) and imported to Adobe Illustrator version 27.1 (2023) for final adjustments and labeling, while (**b, c**) were drawn using Adobe Illustrator version 27.1 (2023).

some intestinal volume increase and potentially stronger stretch reflex and muscle contractions than $L_A$, but less than $S_D$.

Current management of GI motility disorders, including IBS aims to alleviate symptoms rather than addressing the root cause[35,36]. Most constipation drugs work by modifying fluid balance i.e., by decreasing absorption and increasing secretion, thereby increasing the fluid volume in the gut lumen[37,38], but they do not address the primary alterations in GI motility. The limited effectiveness in altering the disease's natural course can be attributed to technical limitations in understanding the complex and simultaneous interactions between ILP, CM and LM contractions and intestinal evacuation. The multi-parameter tracking setup developed in this study overcomes these limitations and gives a quantitative analysis of the complex muscular events governing GI motility and thereby facilitating the development and evaluation of drugs targeting intestine's natural smooth muscle rhythm to achieve proper bowel evacuation.

## Methods

**Experimental mouse model**. Intestinal peristalsis and DD were studied using jejunal segments from 9–12 week old Swiss Albino male mice. The GI electrophysiology of mice and humans is largely similar and therefore justified their use to study GI motility[18]. GI motility relies on intricate interactions between extrinsic and intrinsic neural networks, including the pivotal role

of ICC and smooth muscle cells[39]. Intrinsic neuronal plexuses provide autonomous control over GI function, allowing the intestine to operate independently from extrinsic neuronal inputs[40,41]. This supported the use of isolated intestinal segments in the tissue bath in the present study. All experiments were approved by the University of Florida Institutional Animal Care and Use Committee (IACUC#: 202300000119). Mice were humanely sacrificed by $CO_2$ narcosis followed by cervical dislocation (per American Veterinary Medical Association's Guidelines for the Euthanasia of Animals). Thereafter, a ~4.5 cm jejunal segment located 12 cm proximal to the cecum was identified, dissected and then mounted in a Mayflower tissue bath (Type 813/6), which is a horizontal water-jacketed chamber (Hugo Sachs Elektronik, Harvard Apparatus, USA), held at 37 °C (Fig. 3a–e). Precise length of the intestinal segments was measured after mounting using a Vernier Calliper, and this information was used by the computer program to accurately calculate the edge width and quantify longitudinal movements. Intraluminal perfusion was achieved using a multi-channel roller pump calibrated to deliver Ringer's solution at a steady state rate of ~0.065 ml/min (Figs. 3a and 4). Ringer was maintained at 37 °C in a water-jacketed glass buffer reservoir (73-3440; Hugo Sachs Elektronik/Harvard Apparatus, Germany) by passing the solution through a heat exchanger[42]. The perfusate was passed through a Windkessel (Figs. 3b and 4) to dissipate any pulsatility coming from the roller pumps. A separate multi-channel roller pump was

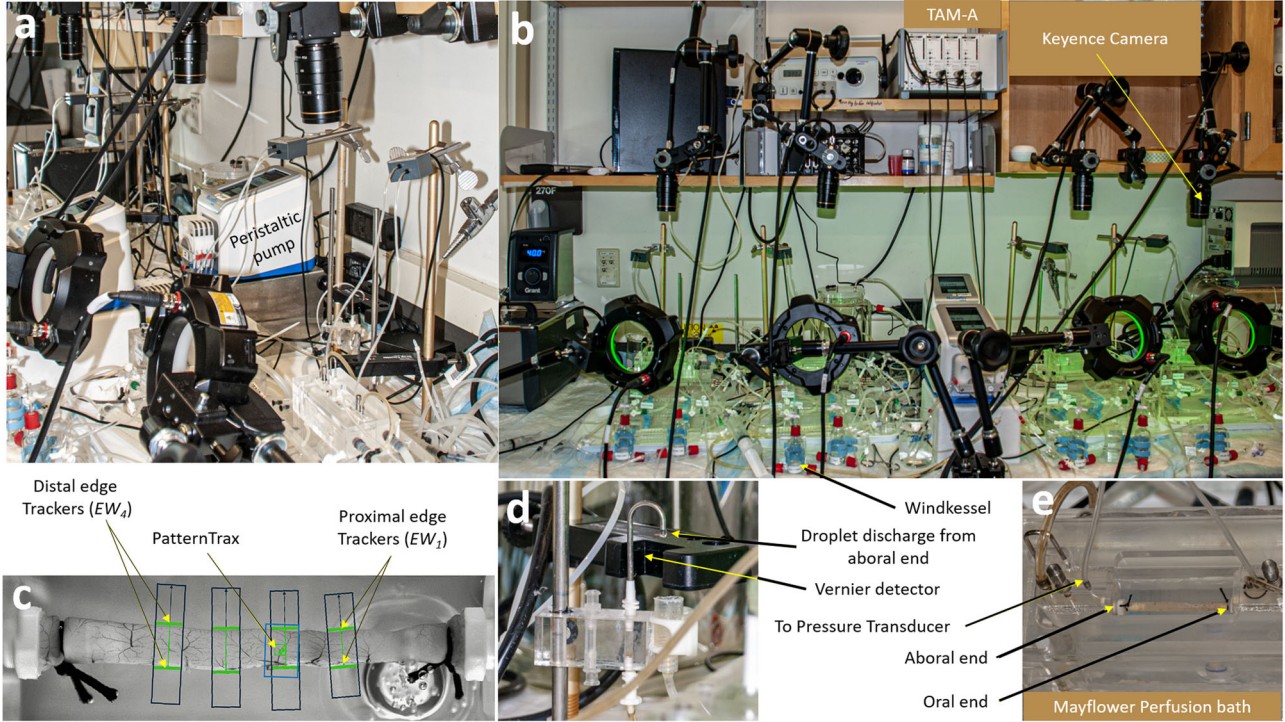

**Fig. 3 The experimental setup for studying peristalsis: a** Lateral view, **b** Front view. PLUGSYS Transducer Amplifier Modules-A (TAM-A, Harvard apparatus, USA), CVX400 series vision system with multi-spectrum LumiTrax light (Keyence, USA) and infrared-LED sensor drop counters with LabQuest Mini interface (Vernier instruments, USA) are used to measure intraluminal pressure, diameter changes, longitudinal movements, and fluid output in intestinal segments. **c** Representative image showing jejunal segment mounted in tissue perfusion bath. Jejunal segment with 4 digital edge width (*EW*) trackers and one PatternTrax placed along the length of the intestine. *EW* trackers track the diameter change with time. PatternTrax measures the longitudinal movement with time using surface characteristics on the intestine such as blood vessels. **d** The luminal perfusate passing through the intestinal segments were made to fall through a specified area of the drop-counter so that the LED light falling on the detector is blocked, resulting in the generation of a digital signal that is then captured by a data collection interface. The data are converted into volume and expressed in ml/min by the data acquisition program. **e** Intestinal segments were perfused in water-jacketed tissue bath (Mayflower, USA). The oral and aboral end of the tissues are connected to the pressure transducers and the pressure recordings obtained were amplified using TAM-A.

used to superfuse the intestinal segment at a rate of 0.2 ml/min also at 37 °C. The Ringer solution for both perfusion and superfusion was continuously bubbled with 95% $O_2$ and 5% $CO_2$ (carbogen) and maintained at pH 7.4 (Fig. 4). Using this setup, the intestinal segments studied stayed viable for ~ 45 min. During this period, the ILP and volume changes were maintained grossly at a steady level. As the tissue deteriorated in the chamber, there were gross reduction in ILP and increase in volume with time. Therefore, most of the studies were limited to the first 30 min. In this setup, the lumen was filled using a peristaltic pump and an afterload was applied to generate an ILP ranging from 0 to 5 $cmH_2O$ column by increasing the height of the luminal efflux tubing (Figs. 3d and 4), as performed in previous studies[43–45]. Each perfusion pressure was maintained for 10 min and the ILP of 1.5 $cmH_2O$ was found to have the maximum amplitude pressure waves maintained for the longest period of time. The afterload help exert the mechanical stretch reflex via the enteric nervous system (ENS) that resides within the wall of the GI tract, including intrinsic primary afferent neurons (IPANs), interneurons, and motor neurons to achieve smooth muscle contraction[45,46]. IPANs are sensory neurons that detect the mechanical, hormonal, or chemical changes in the gut and the interneurons transmit the information to the enteric neural network, while the motor neurons provide the desired muscular activity[41]. This triggers mechanically a cascade of alterations in the CM and LM frequency, and contraction patterns, ultimately resulting in DD at the aboral end with different duration.

**Intraluminal pressure measurements**. ILP plays an important role in the movement of luminal content through the intestine, and was recorded in the past using high resolution manometry and pressure transducers connected to the oral and aboral ends. In the current study, the oral and aboral ends of the tissues were connected to two individual pressure ports at the top of the perfusion bath by using Tygon tubing with an inner diameter of 0.79 mm attached to a differential pressure transducer MPX (type 399/2) that uses a monolithic silicon piezoresistor (Hugo Sachs Elektronik/Harvard Apparatus, Germany). The pressure was calibrated after adjusting the water column using the afterload control (Fig. 4). This setup therefore accurately delivers pressure changes occurring in the intestinal segment (PBTO)[42]. Such a setup does not impede or alter the natural movement of the intestine. The pressure signal from the transducers were amplified using a universal DC bridge amplifier (Transducer Amplifier module (TAM-A) type 705/1 (Hugo Sachs Elektronik/Harvard Apparatus, Germany; Figs. 3b and 4). Four such TAM-A modules were housed within an HSE Plugsys measuring system. The Plugsys apparatus captured signals originating from the isolated intestinal segments mounted in PBTOs, and amplified and continuously recorded every 2 msec, using a Data Acquisition Hardware and Basic data acquisition software (HSE-BDAS; Hugo Sachs Elektronik/Harvard Apparatus, Germany) running on a Windows 10 operating system. A low-pressure afterload of 1.5 $cmH_2O$ was applied using a raised water column to simulate intestinal distention during the passage of the food (Figs. 3e and 4). The pressure waves acquired by the BDAS software were

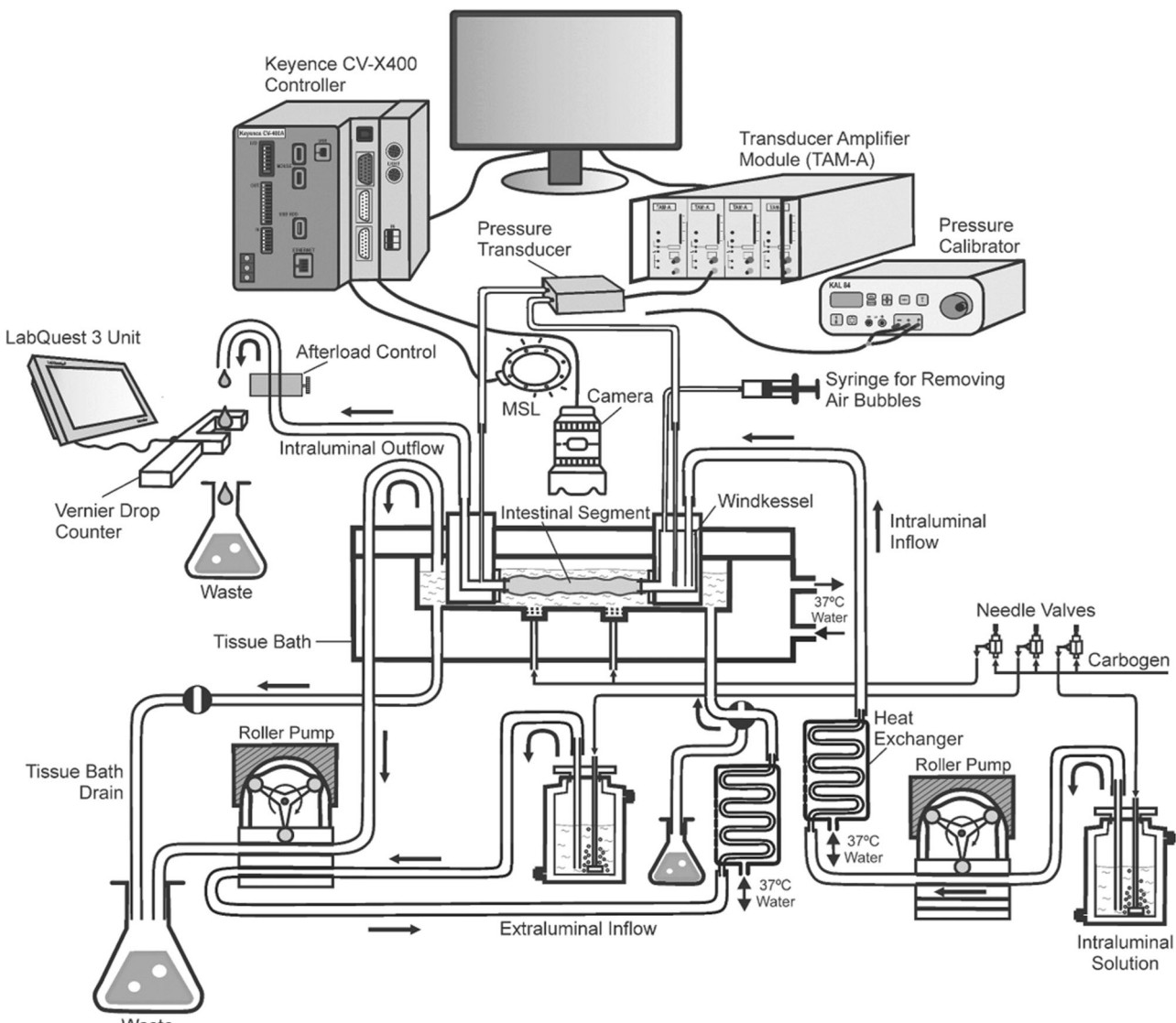

**Fig. 4 Flow diagram showing the setup for studying peristalsis.** The setup involves tissue bath, with units for pressure transducer-amplifier, camera control and droplet discharge sensor-meter. The setup allows for a comprehensive study of intestinal motility capturing movements of the intestinal segment mounted in a perfusion bath using a Keyence camera, multi-spectrum LumiTrax light (MSL) and CV-X400 series controller system. Changes in intraluminal pressure were measured using a pressure transducer and transducer amplifier module (TAM-A) connected to the ports on the tissue bath. A droplet discharge occurring at the aboral end was detected using the infrared-LED sensor drop counter (Vernier Instruments, USA) and metered using LabQuest Mini Interface. Together this setup allows for a comprehensive study of intestinal motility and enhances the understanding of complex biological processes responsible for droplet discharge. The figure was created using Corel Draw, version 21.3.0.755 (2019).

too enormous to manually quantify the number of high amplitude waves and their duration. Therefore, a proprietary program, peristalsis.exe[47] was used to accurately quantify changes in pressure, frequency, amplitude, strength, and duration of contractions. These pressure parameters were later correlated with peristaltic events such as changes in outer diameter, longitudinal movements of the intestine and DD at the aboral end. Predetermined thresholds were set in the program to differentiate low and high amplitude pressure changes, facilitating analysis of large data sets. Our plan going forward was to use this experimental setup to assess segmental differences in ILP, and CM and LM contractions and their corresponding frequency (Fig. 5a–i), with and without various pharmacological interventions intended to mimic GI motility dysfunction in human disease states.

Graphical data obtained using BDAS displayed both low and high amplitude pressure changes (Fig. 5e–g). The frequency of

high amplitude contractions in jejunal segments were determined using "Peristalsis.exe"[47] while low amplitude contractions were assessed manually. The frequencies of both low and high amplitude contractions were similar and not statistically different within each corresponding intestinal segment ($0.69 \pm 0.01$ Hz vs. $0.69 \pm 0.01$ Hz; $p > 0.05$, $n = 192$). However, there was a significant difference in frequency between the jejunal and ileal segments ($0.76 \pm 0.01$ Hz vs. $0.55 \pm 0.01$ Hz; $p < 0.001$, $n = 22$). This aligns with prior research indicating that the frequency progressively decreases in the distal regions of the GI tract compared to the jejunum[30,48]. Given the distinct frequency differences in the jejunum and ileum, indicating the involvement of separate pacemaker groups, these segments were individually analyzed to uncover variations in muscular activities responsible for fluid discharge. The present study specifically concentrates on the jejunal gut segment.

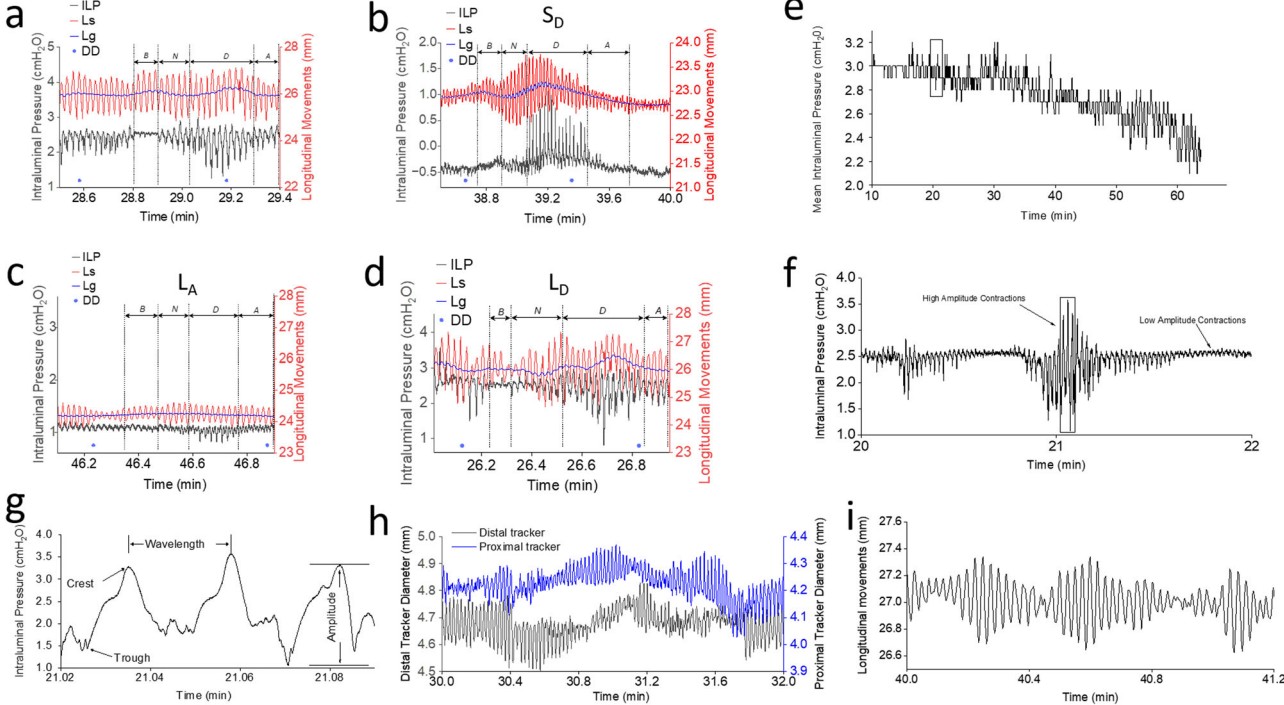

**Fig. 5 Representative tracings for intraluminal pressure, longitudinal and circular muscle contraction, and droplet discharge. a** The intraluminal pressure (ILP) tracings showing separation into four phases based on $Ps$, $Ls$ and/or $Lg$. $Lg$ trace (blue) is obtained by selecting the mean of each $Ls$. "Before-phase" ($B_{Phase}$) started with low $Lg$ and $Ls$, and both progressively increased by the end of the phase. During $B_{Phase}$, low amplitude pressure waves were observed, and this phase was followed by a phase with increased $Ps$, called the "near-phase" or $N_{Phase}$. The $N_{Phase}$ started with a rapid increase in $Ps$, and during this period $Ls$ progressively increased and reached a peak or plateau marking the end of the $N_{Phase}$. The $N_{Phase}$ displayed slightly higher $Ps$ when compared to $B_{Phase}$. The $N_{Phase}$ preceded the high amplitude $Ps$ in "during-phase" ($D_{Phase}$). Both, $Ls$ and $Lg$ progressively increased and reached a peak from the beginning of the $N_{Phase}$. The $D_{Phase}$ is the region of the pressure tracing where the highest amplitude contraction waves were observed and was significantly higher when compared to $B_{Phase}$ and $N_{Phase}$. The $D_{Phase}$ had the longest duration and was followed by "after-phase" ($A_{Phase}$). At the end of the $D_{Phase}$, $Ps$, $Pg$ and $Lg$ returned to the base base levels and marked the beginning for the $A_{Phase}$. Duration and amplitude ($Ps$) for $A_{phase}$ was significantly lower when compared to $D_{Phase}$ but was comparable to that of $B_{Phase}$. **b** Representative tracings for $S_D$. The figure shows a significant increase in longitudinal movements ($Lg$ and $Ls$) from $B_{Phase}$ to $D_{phase}$, during which $Ps$ showed a significant increase. DD occurred in $D_{Phase}$ (blue dot). In $A_{Phase}$, $Lg$ and $Ls$ decreased together with a decrease in $Ps$ and $Pg$. **c** Representative tracings for $L_A$. The amplitude of $Ps$, $Pg$, $Ls$ and $Lg$ are relatively smaller when compared to $S_D$ and $L_D$. $Lg$ and $Ls$ increased from $B_{Phase}$ to $D_{Phase}$ and decreased from $D_{Phase}$ to $A_{Phase}$. DD occurred in $A_{Phase}$ (blue dots). **d** Representative tracings for $L_D$. An increase in $Lg$ was observed from $N_{Phase}$ to $D_{Phase}$ but was significantly lower than $S_D$ drops. Duration for $L_A$ and $L_D$ were significantly higher when compared to $S_D$. The decrease in $Lg$ during the $A_{phase}$ in $L_D$ was significantly lower than that observed in $S_D$ drops. DD occurred in $D_{Phase}$. At the end of $A_{Phase}$ $Lg$ decreased to the levels observed in $B_{phase}$. DD occurred in $D_{Phase}$ (blue dot). Decrease in $Lg$ in $A_{phase}$ was observed in all DD groups. **e** Representative pressure trace showing trend over time. ILP maintained steady levels for up to 40 min and thereafter, ILP decreased significantly. Therefore, all subsequent studies were performed within the first 30 min. **f** Zoomed view from (**e**) (20–22 min) showing low and high amplitude pressure waves ($Ps$) with gross pressure changes ($Pg$). Pressure tracings show high amplitude waves flanked by small rise and fall in pressure strength. **g** Zoomed view from (**f**) (21.02–21.1 min) showing crest, trough, amplitude, and wavelength. **h** Representative traces showing the diameter changes in proximal ($EW_1$) and distal ($EW_4$) tracker with time. Lower values represent decrease in diameter or circular muscle contraction. **i** Representative trace showing low and high amplitude oscillations ($Ls$) along the gross longitudinal muscle contraction ($Lg$). The $Lg$ tracings recorded using PatternTrax are read from left to right, thereby increased values suggest contraction of longitudinal muscles at the proximal end while decreased values suggest contraction at the distal end.

ILP recordings between tissues were compared between experiments using "gross pressure" ($Pg$) and "pressure strength" ($Ps$). In the detailed pressure recording window of BDAS, we observed multiple high-amplitude pressure waves occurring at irregular intervals (Fig. 5e–g). However, it was unclear how these waves affected $Pg$. $Ps$ of these waves was calculated as the algebraic sum of individual pressure waves amplitudes (measured from crest to trough) divided by the number of waves. While $Pg$ in the intestinal segments studied were calculated from the measured ILP recordings (Fig. 5e), and mathematically represented as $Pg$ = the sum of pressure recordings taken over the study period divided by the number of recordings taken over the study period. Changes in $Pg$ between phases measured the magnitude of pressure changes without considering the oscillations associated with ILP waves.

**Analysis of edge width, volume, longitudinal movement and contraction type.** Spatiotemporal mapping has been the most widely used technique to track CM and LM contractions. In this study, to monitor and quantify two-dimensional edge movements for CM and LM contractions in intestinal segments in real-time, we employed an innovative approach utilizing a CV-X400 series vision system. This setup comprised an ultra-high-speed camera (LumiTrax™), multi-spectrum lighting with eight color LEDs, and a dedicated control circuit (Keyence, USA) (Figs. 3a, b and 4). LumiTrax™ employs a novel imaging technique in which the lighting direction and color were automatically synchronized with the camera through machine learning to collect data and quantify real-time changes in both LM (length) and CM contraction (diameter) amplitudes. Multiple images with 24 image enhancement filters and lighting from various directions are taken and

analyzed to detect patterns, such as blood vessels or mesentery, on the intestinal surface to optimize real-time visualization and tracking during intestinal movements, even when their orientation changes (Fig. 3c). Build-in "Auto-Teach Inspection Tools" allows the camera to "learn" and identify variations and differences in the pattern that may occur with intestinal contraction in real-time, ensuring stable tracking to obtain quantifiable data acquisition of diameter and longitudinal movements. This capability enables real-time tracking with less noise during data capture. The camera operates with preset configurations, reducing setup time, thus enhancing tissue viability. Both longitudinal and circular movements are captured every 50 msec.

Longitudinal movements are tracked and recorded from the aboral end. The gross longitudinal movements ($Lg$) tracked using "PatternTrax" helped determine whether LM contractions were oriented toward the oral or aboral end of the intestine. These movements were recorded as a gross shift along the $Y$-axis ($Lg$) over time (Figs. 5i and 6a–c) and are calculated by subtracting the mean longitudinal position of each phase from the previous phase. Increased $Lg$ values indicate LM contractions toward the oral end, while decreased values suggest gross LM movement toward the aboral end. Due to slight differences in the length of the intestinal segment studied and the position of the PatternTrax on the intestine, the baseline $Lg$ levels were never exactly the same. Therefore, we normalized the values by subtracting the smallest value from all longitudinal recordings to obtain the absolute distance moved, facilitating comparisons between experiments or different phases of DD. These values reflect the strength of the LM contractions. By comprehensively studying the coordinated roles of LM movements, CM movements, and their influence on ILP changes, we can bridge the existing knowledge gap and gain a deeper understanding of how these factors collectively contribute to the propulsion of luminal contents. LM tracings exhibited both low and high amplitude oscillations with amplitude representing the strength of LM contraction, denoted as $Ls$, calculated as the algebraic sum of the amplitudes of individual longitudinal waves divided by the number of waves. To understand the role of $Ls$ in determining the duration of DD and how it interacts with $Lg$, $Ls$ was compared across different phases and DD groups (Fig. 5b–d).

For diameter tracking, we selected opposite edges of the intestine at four regions along its length as edge width ($EW$) trackers using the CV-X400 series machine vision system. These EW trackers measured changes in the distance between opposite edges during contraction generating a graphical representation of diameter changes. All four pairs of EW trackers were used continuously along the sides of intestinal segment, with adjacent trackers spaced apart by approximately the length of one edge tracker. Contraction was indicated by a decrease in EW (diameter), while relaxation was indicated by an increase. The diameter measurement from the four EW trackers changed with changes in fluid volume in the intestine. The analysis of the gross movement of EW trackers ($EWg$) depicts the overall diameter change, which is calculated by subtracting the mean diameter in one phase from that of the previous phase. The amplitude of EW contractions at any given time represents the strength of EW contractions ($EWs$). To ensure that the changes recorded by "GutCode" accurately represented the actual variations in diameter in each of the EW trackers, the magnitude of contraction in each phase was manually measured. The data obtained through "GutCode"[49] aligned well with the gross movement of the proximal edge width tracker ($EW_1g$). The gross movement of edge width tracker contraction waves was calculated by dividing the sum of all the crests by the duration of the time period studied, and this measured the magnitude of the contraction. $EW_1g$ traces showed low and high amplitude

oscillations, and the amplitude of the oscillation was considered as the strength of the contractions, and is denoted as $EW_1s$. To determine the role of $EW_1s$ in the duration of DD, $EW_1s$ was compared between different DD groups.

The propagation of waves along the intestinal wall was captured using four pairs of $EW$ trackers. The propagation of contraction waves along the intestinal wall and the development of anterograde, retrograde, and segmental contractions play a crucial role in intestinal motility and the dynamics of DD (Fig. 6a–e). The proximal tracker contraction with a distal tracker relaxation resulted in anterograde propulsion while a distal tracker contraction with a proximal tracker relaxation generated retrograde propulsion. When both proximal and distal trackers contracted or relaxed simultaneously, the contraction waves neither proceeded orally nor aborally, triggering segmental contraction (Fig. 6d, e).

The changes in EW trackers were used to calculate real time changes in intestinal volume ($V$) using the formula $V = \pi \sum_{i=1}^{4} hi \frac{d_i^2}{4}$, where $d1$, $d2$, $d3$, and $d4$ represent the diameters recorded by corresponding EW trackers, and $h1$, $h2$, $h3$, and $h4$ represent the length of the intestine measured by the trackers, including half the gap between trackers on each side of the gut segment. The formula for volume calculation was integrated into the proprietary program "GutCode"[49] which automated and expedited the computation of diameter and volume changes from EW tracings. However, it's important to note that this volume calculation represents volume changes and could be influenced by the longitudinal and circular muscle contractions, as it was not possible to separate the contributions of circular and longitudinal muscle contractions in our method. The volume calculation helps us understand how luminal filling or emptying influenced DD and ILP changes, or CM and LM movements. Volume tracings showed low and high amplitude contraction waves, and the amplitude of these waves was described as volume strength ($Vs$) and the gross volume movement was described as $Vg$.

**Quantification of anterograde, retrograde and segmental contractions.** We utilized "GutCode"[49] for the analysis of diameter data obtained from four EW trackers positioned along the oral-to-aboral axis of intestinal segments. Contractions were categorized into anterograde, retrograde, and segmental based on relative movements of adjacent trackers. Anterograde was noted when the proximal tracker detected contraction followed by relaxation detected by the distal tracker. Retrograde was identified when the proximal tracker noted relaxation followed by contraction detected by the distal tracker.

Segmental contractions were recognized when both proximal and distal trackers detected contractions simultaneously. To quantify contraction strength, we divided the sum of the amplitudes by time, assigning positive values to anterograde, negative values to retrograde, and segmental values based on their overall inclination toward anterograde or retrograde with a positive or a negative sign.

Since "GutCode"[49] processed data obtained at a 50 msec intervals, and contraction-relaxation cycle occur at 0.7 Hz, where each wave takes ~1.4 s to complete, only minor shifts were analyzed in anterograde, retrograde, or segmental contractions between $EW$ trackers for each data point. In addition, the slow waves generating propulsive movements can instantaneously change direction and speed of propagation[50]. Therefore, to determine the overall contraction direction, we introduced "net amplitude", accounting for small amplitude shifts among all four trackers. A "net amplitude" > 0 indicated anterograde, while <0 indicated retrograde contractions. "Net amplitude" allowed for comparisons between different discharge groups and contraction

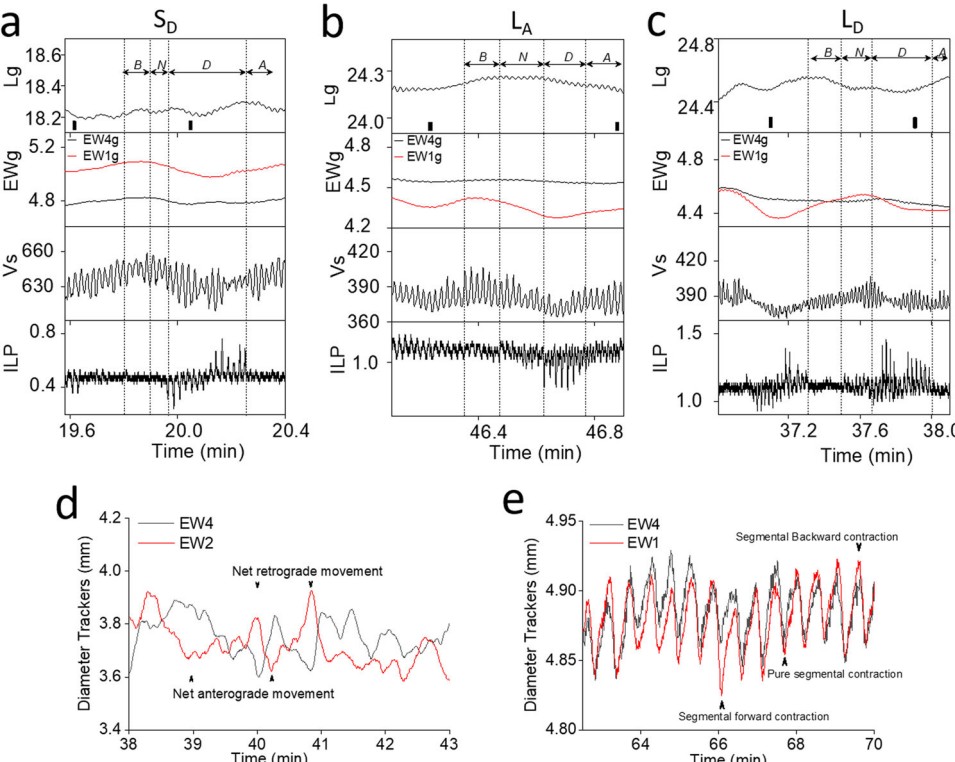

**Fig. 6 Panel graphs showing traces of factors responsible for droplet discharge duration and segmentation: a** $S_D$ discharge: in $B_{Phase}$ to $N_{Phase}$, longitudinal muscles showed maximum values, $EW_1$ and $EW_4$ begins to contract with maximum reduction occurring in the $D_{Phase}$. Longitudinal muscle contraction at proximal end occurred before circular muscle began to contract. $EW_1g$ showed greater contraction when compared to $EW_4g$. Correspondingly, maximum increase in volume was seen in $B_{Phase}$ and thereafter, the volume began to decrease with maximum reduction in the middle of $D_{Phase}$. $Lg$, $EWg$ and $Vg$ showed reduction in the $D_{Phase}$, where $Ps$ was increased. $Lg$ increased from the middle of $D_{Phase}$ and reached a maximum toward the early part of $A_{Phase}$ while, $EWg$ and $Vg$ continued to increase through the $A_{Phase}$. $Ps$ decreased during the $A_{Phase}$. **b** $L_A$ discharge: longitudinal muscles showed an increase from $B_{Phase}$ to $D_{Phase}$ and thereafter decreased from the beginning of $D_{Phase}$ and continued into the $A_{Phase}$. $EW_1$ started to contract in $B_{Phase}$, $N_{Phase}$ and early $D_{Phase}$, while longitudinal muscle remained in the contracted state at the proximal end. $EW_1g$ showed a marginal increase in $D_{Phase}$ and $A_{Phase}$. Unlike $EW_1g$, $EW_4g$ contractions were minimal. Volume changes are minimal compared to $S_D$. An increase in volume was observed from $N_{Phase}$ to $D_{Phase}$, followed by a decrease in $A_{Phase}$. **c** $L_D$ discharge: longitudinal muscle contraction showed a decrease in $B_{Phase}$ to $N_{Phase}$, unlike $S_D$ and $L_A$. $EW_1g$ values increased from $B_{Phase}$ to $D_{Phase}$, suggesting continued filling through $B_{Phase}$ and $N_{Phase}$, and began to empty only in the $D_{Phase}$. Unlike $S_D$ and $L_A$, in $L_D$, the lumen continued to fill through $B_{Phase}$ and $N_{Phase}$. $EW_4g$ continued to decrease from $B_{Phase}$ to $A_{Phase}$ with maximum decrease occurring in $B_{Phase}$ to $N_{Phase}$. Volume decreased in $D_{Phase}$ and did not return to base levels suggesting incomplete emptying. **d** Diameter changes between $EW_1$ and $EW_4$ depicting anterograde and retrograde contraction analyzed using "GutCode". $EW_1$ contraction with $EW_4$ relaxation causes anterograde movement (upward arrow), and $EW_1$ contraction with $EW_4$ relaxation causes retrograde contraction (downward arrow). **e** Diameter changes between $EW_1$ and $EW_4$ depicting segmental contractions analysed by "GutCode". Simultaneous contraction or relaxation of $EW_1$ and $EW_4$ causes pure segmental contraction. A small increase in diameter of $EW_4$ compared to $EW_1$ causes segmental forward, and a small decrease in diameter of $EW_4$ compared to $EW_1$ causes segmental backward contractions.

phases. These recordings provided insights into various contractions, tabulating their frequency, strength, duration, amplitude, retrograde, segmental, and anterograde characteristics, and their impact on intestinal DD. This tabulation facilitated easy comparisons between studied tissue segments (Figs. 1a–d, 5a–i, 6a–e and Tables 1 and 2).

**Experimental analysis of droplet discharge**. The outflow from the luminal perfusate that constituted a DD was directed to pass through a $1.3 \times 3.7$ cm opening of Vernier Drop Counter, equipped with an infrared (890 nm) LED emitter at one end, and a detector at the opposite end. When a drop of perfusate obstructed the infrared beam between the emitter and detector, a digital signal was transmitted to the LabQuest®3 system. These drops were then converted into microliters (μl) using a calibration chart within the LabQuest®3 program (Figs. 3d and 4). This method allowed for precise and automated measurement of the

time and volume of each DD event occurring at the aboral end in the studied intestinal segments. Thus, DD duration was assessed as the interval between two DD.

Rate of fluid flow through the tubing and/or intestinal segment was represented as "Flow rate". The flow rate was calculated by dividing the volume of a drop obtained at the aboral end by the time elapsed between DD events, and was represented as ml/min. The flow rate exhibited slight variations between experiments due to minor differences in flow arising from the peristaltic pump and tubing employed. The average flow rate was 0.065 ml/min. The flow rate through the intestine was therefore determined by running the perfusate through a Tygon tubing in lieu of the intestinal segment, both before and after the experiment, and the mean was represented as the average flow rate. The flow through the intestinal segment or the Tygon tubing was achieved using a peristaltic pump (Figs. 3a and 4). Accurate determination of DD event duration is crucial for precise muscular event evaluation. To

account for slight variations in flow duration, we normalized the experimental duration (D) for each DD event while the intestine was positioned in the Mayflower bath by dividing D by the standard duration (Ds) when the perfusate passed through a Tygon tubing. Consequently, we utilized the normalization method D/Ds to mitigate potential variations in the flow rate introduced by the pump and tubing during the experiment. When the D/Ds ratio approached one, it indicated minimal variations.

The total inflow and outflow volume over a 15-min period was calculated, and their difference determined the secretory or absorptive nature of the studied intestinal segment. An increase in outflow volume indicated secretion, whereas a decrease suggested absorption. The variations in D/Ds were compared to other muscular events such as changes in ILP, longitudinal movements, diameter (i.e., $EW$ contractions). This analysis provided insights into the mechanisms governing fluid flow and discharge at the aboral end of the gastrointestinal tract.

Since pressure waves were a reflection of CM and /or LM contractions, they were effectively employed as a reference to deduce DD, and assess all muscular activities. When categorizing DD duration, it was observed that DDs could be either short or long in duration. These distinctive durations were closely associated with various muscular parameters and amplitudes of the pressure waves. Therefore, when DDs were sorted by duration, discharges were found to occur either in $D_{Phase}$ or $A_{Phase}$ (Fig. 5a). Among the 48 DDs studied, the shortest duration DDs primarily occurred in $D_{Phase}$, making up 37.5% of total discharges and constituting the $S_D$ group (Fig. 5b). Majority of DDs (39.6%) with longer durations occurred in $A_{Phase}$, and were represented as the $L_A$ group (Fig. 5c). Additionally, 16.7% of DDs occurring in $D_{Phase}$ had longer durations, classified as the $L_D$ group (Fig. 5d). Three DDs, which occurred in $A_{Phase}$ but had shorter durations compared to $L_A$, and one discharge occurring in both $D_{Phase}$ and $A_{Phase}$ with a long duration, were excluded from further analysis. Thus, based on duration and D/Ds, we identified three discharge groups: $S_D$, $L_A$, and $L_D$.

To gain further insights, we divided the pressure contractions during each DD into four distinct phases (Fig. 5a) based on the $Ps$. Pressure levels ranging from 0.03 to 0.2 cmH₂O were considered low amplitude pressure waves, while pressures exceeding 0.2 cmH₂O indicated high amplitude contractions. These high amplitude pressure waves were further categorized based on $Ps$, $Ls$, and/or $Lg$. These parameters exhibited significant changes during these high amplitude pressure waves:

- The $B_{Phase}$ featured low $Ls$ and $Lg$, both of which gradually increased throughout this phase. During the $B_{Phase}$, low amplitude pressure waves were observed.
- The $N_{Phase}$ began with a rapid $Ps$ increase following $B_{Phase}$. During this phase, $Ls$ progressively increased and reached a peak or plateau which marked the end of the $N_{Phase}$. $Lg$ did not increase as much as in $B_{Phase}$ or in some instances it showed a small decrease. The increase in $Lg$ from $B_{Phase}$ to $N_{Phase}$ and $N_{Phase}$ to $D_{Phase}$ suggests LM movement and contraction at the oral end. Conversely, the decrease in $Lg$ in $A_{Phase}$ suggests LM movement and contraction at the aboral end.
- The $D_{Phase}$ followed the $N_{Phase}$ and was characterized by progressively increasing $Ls$ and $Lg$, both reaching a peak. This phase exhibited the highest amplitude pressure contractions. $D_{Phase}$ showed pressure tracings with the highest amplitude contraction waves and was significantly higher when compared $N_{Phase}$ ($p < 0.001$, $n = 48$). $D_{Phase}$ had the following characteristics: (1) Increase in $Ps$ and reaching a plateau that marks the end of $D_{Phase}$; (2) Increase in $Ls$ and $Lg$ followed by a decrease, which

paralleled changes in $Ps$. Thus, at the end of the $D_{Phase}$, $Ps$, $Pg$ and $Lg$ return to baseline levels, which marked the beginning of the $A_{Phase}$ (Fig. 5a).
- The $A_{Phase}$ succeeded the $D_{Phase}$. It was shorter in duration and featured lower amplitude pressure contractions compared to $D_{Phase}$, resembling $B_{Phase}$ in characteristics. The total duration ($20.1 \pm 0.1\%$ vs. $45.7 \pm 0.8\%$) and amplitude ($0.23 \pm 0.02$ cmH₂O vs. $0.66 \pm 0.07$ cmH₂O; $p < 0.001$, $n = 48$) for $A_{Phase}$ was significantly lower than $D_{Phase}$. $A_{Phase}$ was comparable to $B_{Phase}$ (Fig. 5a).

These phases provided valuable information about the relative movement and contraction of LM and CM, as well as their timing. This enhanced our understanding of the muscular activity that initiates or precedes each contraction.

**Statistics**. Statistical analysis was conducted using OriginPro 9.9 (2022), and the different phases were compared within each discharge group. The data were presented as mean ± SEM, and the range of the data sets for DD was shown. Normality of the data sets was evaluated with Shapiro–Wilk test. One-way ANOVA was employed, followed by post hoc Bonferroni test to determine significant differences of data between the four phases. Kruskal–Wallis test was used for overall comparison of the discharge groups, and post hoc Mann–Whitney test was utilized to compare discharge groups between experiments. A significance level ($p$) was set as <0.05.

**Reporting summary**. Further information on research design is available in the Nature Portfolio Reporting Summary linked to this article.

## Data availability
All raw data that support this study are available as Supplementary Data and software files in https://zenodo.org/[51].

## Code availability
The software applications used to extract and calculate the data points are available as Supplementary Data and software files in https://zenodo.org/[47,49].

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

## Acknowledgements

This investigation was funded by R01 grants # DK074867 and # DK109717 from the National Institute of Diabetes and Digestive and Kidney Diseases (NIDDK), and the Entrinsic Bioscience Research Fund (# P0196836). We would also like to acknowledge Sheldon Xu, Damiano Angoli and Ahrad Nathan for assistance with instrumentation.

## Author contributions

S.V. conceived the project and methodology, and S.V. and J.F.C. contributed to the funding. A.G. conceived the project, and S.V., A.G. and A.S. developed the setup and performed the experiments. A.V.K. and A.H. developed the software applications. A.S., B.A.P. and K.S.K. performed the computational analyses. S.V., A.S. and B.A.P. wrote the manuscript. B.A.P. created Fig. 2a, while S.V. created Fig. 2b, c, and A.G. created Fig. 4. S.V., A.G., N.F. and J.F.C. performed data interpretation and edited the manuscript. All authors have given their approval to the final version of the manuscript.

## Competing interests

The research was partially funded by Entrinsic Bioscience. S.V. is the Chief Science Officer of and has shares in Entrinsic Bioscience. K.S.K., A.H. and N.F. are consultants for Entrinsic Bioscience. PCT/US2023/017225: Peristaltic Propulsion Device and System and Methods of Use Thereof: Inventors: S.V., A.S., B.A.P., K.S.K., A.G., A.V.K. and A.H. US No. 18/194,632: Peristaltic Propulsion Device and System and Methods of Use Thereof: Inventors: S.V., A.S., B.A.P., K.S.K., A.G., A.V.K. and A.H. US Provisional: No. 63/493,748: Systems and Methods for Analyzing Intestinal Peristalsis. US Provisional: No. 63/599,812: Methods for Use of Peristaltic Propulsion System. J.F.C. does not have competing interests.
