## [Peer Review File · Communications Biology]

Reviewers' comments:

Reviewer #1 (Remarks to the Author):

REVIEW REPORT

Authors have performed the study relating to Advancing Peristalsis Deciphering in Mouse Small Intestine by Multi-Parameter Tracking. It appears that the authors have not gone through sufficient prior art work in the area. The draft lacks proper documentation of the literature, methodology, tabulation, analysis and discussions. Although authors have made a good attempt to replicate the data already studied in literature, they were unable to specifically highlight the research gaps and the scope of the study. The overall presentation needs to be heavily improved. Key works are not cited in the literature/ discussed.

Probably, this sentence needs revision, "Currently, assessing gastrointestinal motility lacks simultaneous evaluation of intraluminal pressure (ILP), spatiotemporal mapping for circular muscle (CM) and longitudinal muscle (LM) contraction and lumen emptying." In humans CM and LM studies with manometry has been made, refer to Nicosia Work (esophagus).

This sentence, "Poor understanding of the underlying mechanisms and complex interplay of factors involved often render GI motility disorders as intractable, chronic, and prolonged diseases." also needs revision, as there are literature studies including computational works.

In page 37, author may use a block diagram to show the complete experimental set-up. As it remains confusing to identify from the experiment set-up shown. This will improve clarity.

In page 38, cmH₂O may be changed to mmHg, and plots as well, as per the preference of author's. Since pressure indication in mmHg is more used in literature, such as for HRM, may be considered.

Fig. nos. are not appearing in the draft, author may check this.

In page 40, a clear distinction on the type of motility may be provided, as it is not clear on what wall movements are prevalent as shown in the figure. Wall motion may be illustrated for clarity.

No details on Gut Code, a flowchart or details may be provided with the algorithm on what and how calculations were made.

In "Longitudinal movements are tracked and recorded from the aboral end", illustrated may be detailed with tracker images alongside one such movement plots, such as gross longitudinal 172 movements (Lg).

The technical definitions are missing for Lg, Ls, DD and others as well. It appears to be qualitative or confusing. (Page 9). The same applied for Edge Width, Volume, Longitudinal Movement and Contraction Type (page 8) and others.

The term DD appears to be misleading, author may reconsider changing it to a more suitable term.

Clear definition to be included for flow rate. In "Flow rate was calculated by dividing the drop's volume by the time elapsed between DD events.", it is ambiguous. A formula may be included for this, and for

other parameters considered in the study.

I would suggest to introduce a separate section, clarifying terms, nomenclature and definitions.

The para, "Current management of GI motility disorders, including IBS aims to alleviate symptoms rather than addressing the root cause^{38,39}. Most constipation drugs work by modifying fluid balance in the gut^{40,41}." this seems to be diverging from the central theme of the work. With discussions highly digressing and not channelling towards a meaningful deliberation of the results with literature data.

There is insufficient literature cited both in the introduction as well as in the discussion sections.

Whereas the central theme of the work is imaging analysis of the intestinal segment, previous work was not discussed at all.

Justification of the study may be given with due preference to the physiology.

Dr. Ravi Kant Avvari
NIT Rourkela

Reviewer #2 (Remarks to the Author):

This manuscript delineates the design and development of a comprehensive system dedicated to the multi-faceted monitoring of small intestine motility. By integrating advanced techniques such as multi-spectrum imaging, pressure sensors, and Vernier Drop counters, the system showcases its capability to extract detailed data. The insights garnered from this device hold significant promise for propelling gastrointestinal research forward. While the manuscript is commendably articulated, I would like to offer the following constructive feedback:

1. In Fig. 1, it would be beneficial to label all the primary devices explicitly. Additionally, within Fig. 1B, please indicate the positions of the sub-components depicted in Figs. C-E for clarity.
2. The manuscript mentions the machine learning capabilities of the camera but does not elucidate how this enhances image quality. A more detailed explanation would be appreciated.
3. Can the authors provide assurance that the incorporation of pressure sensors at both tissue extremities does not impede or alter the natural movement of the intestine?
4. It would be informative to know the viable duration for which the intestine remains functional post-dissection.
5. On Page 3, references 7-10 seem to overlook some of the recent advancements in intestinal imaging. I recommend considering the inclusion of the following studies:

Wang, D., et al. (2021). Trans-illumination intestine projection imaging of intestinal motility in mice. *Nature Communications*, 12(1), 1682.

Boquet-Pujadas, A., et al. (2022). 4D live imaging and computational modeling of a functional gut-on-a-chip evaluate how peristalsis facilitates enteric pathogen invasion. *Science Advances*, 8(42),

eabo5767.

6. The manuscript references a "proprietary program, peristalsis.exe" but offers limited details about its innovative features. Could you elucidate how the "predetermined thresholds" were established? Additionally, it would be beneficial to know if this program has undergone validation against traditional methodologies.

Responses to the Reviewers' comments

Reviewer #1 (Remarks to the Author):

REVIEW REPORT

Authors have performed the study relating to Advancing Peristalsis Deciphering in Mouse Small Intestine by Multi-Parameter Tracking. It appears that the authors have not gone through sufficient prior art work in the area. The draft lacks proper documentation of the literature, methodology, tabulation, analysis and discussions. Although authors have made a good attempt to replicate the data already studied in literature, they were unable to specifically highlight the research gaps and the scope of the study. The overall presentation needs to be heavily improved. Key works are not cited in the literature/ discussed.

Reviewer 1

1. Probably, this sentence needs revision, ""Currently, assessing gastrointestinal motility lacks simultaneous evaluation of intraluminal pressure (ILP), spatiotemporal mapping for circular muscle (CM) and longitudinal muscle (LM) contraction and lumen emptying." In humans CM and LM studies with manometry has been made, refer to Nicosia Work (esophagus).

Response: We agree with the reviewer that the Nicosia's Work should be included as a reference. However, the novelty of the current setup and the study is that changes in ILP, CM contraction and LM contraction are correlated to droplet discharge in mouse jejunum and are all quantitative. Although Nicosia work studied coordinated movement of ILP, CM contraction and LM contraction for bolus movement, it is limited to esophagus and is not quantitative to an extent where differences in the strength of the contraction and frequencies can be ascertained as droplet discharge occurs with shorter or longer duration.

2. This sentence, "Poor understanding of the underlying mechanisms and complex interplay of factors involved often render GI motility disorders as intractable, chronic, and prolonged diseases." also needs revision, as there are literature studies including computational works.

Response: The sentence is rewritten as 'Although the underlying mechanisms and complex interplay of factors involved in GI motility have been explained previously by multiple studies, the management of GI motility disorders is still complicated by an incomplete understanding of the precise mechanisms coordinating GI smooth muscle activity, which are necessary for proper bowel emptying. The references citing the articles have been included.

3. In page 37, author may use a block diagram to show the complete experimental set-up. As it remains confusing to identify from the experiment set-up shown. This will improve clarity.

Response: Thank you for the suggestion. A block diagram is now included.

4. In page 38, cmH₂O may be changed to mmHg, and plots as well, as per the preference of author's. Since pressure indication in mmHg is more used in literature, such as for HRM, may be considered.

Response: Centimeter of water (cmH₂O) being a smaller unit of pressure than a millimeter of mercury (mmHg), it can give better representation of the small pressure changes. Since the current study uses jejunum from mouse, the ILP changes are small. Also, there are enough articles where ILP has been represented in cmH₂O and they are referenced in the text.

5. Fig. nos. are not appearing in the draft, author may check this.

Response: Figure number and table numbers are now included in the draft.

6. In page 40, a clear distinction on the type of motility may be provided, as it is not clear on what wall movements are prevalent as shown in the figure. Wall motion may be illustrated for clarity.

Response: Thank you for the suggestion. The legends have been modified as follows to convey the motility type. **(B)** Figure showing longitudinal muscle contraction with gathering of circular muscles rings (muscle fibers) towards the distal end that results in contraction with narrowing of the lumen at the aboral end, favouring fluid-filling. Fluid filling leads to ballooning proximal to the contracted region (bold arrows). Intestinal flow is reduced during the filling stage (thin arrow). Arrows in the lumen depicts more retrograde movement. **(C)** Figure showing longitudinal muscle contraction (parallel lines) with gathering of circular muscles rings (muscle fibers) and narrowing of the lumen at the proximal end that then favours droplet discharge (bold arrows). Arrows in the lumen suggest more anterograde movement.

7. No details on Gut Code, a flowchart or details may be provided with the algorithm on what and how calculations were made.

Response: The program was used to expedite the computation of diameter and volume changes, longitudinal muscle contraction, anterograde, retrograde, and segmental contraction, and net flow. The calculations used in the GutCode is described in the methods section. The text has been modified to bring clarity to the above statement as follows. A custom software program, named GutCode, was developed to compute volume changes and the strength of contractions. The GutCode cleans and smoothens the data using a moving averages algorithm. The gross movement of the edge trackers were estimated using this smoothed data. The gross movement of edge tracker contraction waves was calculated using raw data by dividing the sum of all the crests by the study period, which measured the magnitude of the contraction. The gross movement computed from both raw and smoothed data matched. To ensure the accuracy of the volume and contraction computed by 'GutCode', these quantities were manually computed in each phase and were found to be aligned.

The formula for volume calculation was integrated into the proprietary program 'GutCode', which automates and expedites the computation of diameter and volume changes from EW tracings. However, it's important to note that the volume changes could be influenced by longitudinal and /or circular muscle contractions. In our method, it was not possible to separate the contributions of these contractions.

We utilized 'GutCode' to analyze the diameter data obtained from four EW trackers positioned along the oral-to-aboral axis of the intestinal segments. Contractions were categorized as anterograde, retrograde, or segmental based on the relative movements of adjacent trackers.

8. In "Longitudinal movements are tracked and recorded from the aboral end", illustrated may be detailed with tracker images alongside one such movement plots, such as gross longitudinal 172 movements (Lg).

Response: The signal or image captured using Keyence camera occurs every 50 millisecond, therefore, for every second of recording, there are 20 images captured and thus making it nearly impossible to show the tracker images alongside the muscle movement plots. However, the graphs 2I and 3A-C clearly show tracking of longitudinal muscle movements at a capture rate of one data point every 50 ms for the period analyzed.

9. The technical definitions are missing for Lg, Ls, DD and others as well. It appears to be qualitative or confusing. (Page 9). The same applied for Edge Width, Volume, Longitudinal Movement and Contraction Type (page 8) and others.

Response: Technical definitions for gross movement (g) and strength of contractions (s) for pressure, longitudinal movement, edge width contractions and volume has been corrected and included with the text in the method section.

10. The term DD appears to be misleading, author may reconsider changing it to a more suitable term.

Response: DD is used as an abbreviation to for droplet discharge.

11. Clear definition to be included for flow rate. In "Flow rate was calculated by dividing the drop's volume by the time elapsed between DD events.", it is ambiguous. A formula may be included for this, and for other parameters considered in the study.

Response: Thank you for the suggestion and changes have been made to the text to bring clarity to the flow rate.

12. I would suggest to introduce a separate section, clarifying terms, nomenclature and definitions.

Response: A separate table with abbreviations and definitions of terms used is now included as Table 1.

13. The para, "Current management of GI motility disorders, including IBS aims to alleviate symptoms rather than addressing the root cause. Most constipation drugs work by modifying fluid balance in the gut." this seems to be diverging from the central theme of the work. With discussions highly digressing and not channeling towards a meaningful deliberation of the results with literature data.

Response: Thank you for the suggestion and changes have been made to the text and appropriate references added.

14. There is insufficient literature cited both in the introduction as well as in the discussion sections.

Response: Additional references are now included to reflect the previous work involving image analysis of the intestinal segments.

15. Whereas the central theme of the work is imaging analysis of the intestinal segment, previous work was not discussed at all.

Response: Additional references are now included to reflect the previous work involving image analysis of the intestinal segments.

16. Justification of the study may be given with due preference to the physiology.

Response: The physiological relevance of the study is clearly mentioned in the introduction and discussion.

Reviewer #2 (Remarks to the Author)

This manuscript delineates the design and development of a comprehensive system dedicated to the multi-faceted monitoring of small intestine motility. By integrating advanced techniques such as multi-spectrum imaging, pressure sensors, and Vernier Drop counters, the system showcases its capability to extract detailed data. The insights garnered from this device hold significant promise for propelling gastrointestinal research forward. While the manuscript is commendably articulated, I would like to offer the following constructive feedback:

1. In Fig. 1, it would be beneficial to label all the primary devices explicitly. Additionally, within Fig. 1B, please indicate the positions of the sub-components depicted in Figs. C-E for clarity.

Response: Thank you for the excellent suggestion and to bring clarity and to better understand Fig.1, a flow chart is now included to explicitly show all the devices and positions of the sub-components depicted in Figs. C-E for clarity.

2. The manuscript mentions the machine learning capabilities of the camera but does not elucidate how this enhances image quality. A more detailed explanation would be appreciated.

Response: The text has been modified to bring clarity to the machine learning capability of the camera as shown below. LumiTrax™ employs a novel imaging technique in which the lighting direction and color were automatically synchronized with the camera through machine learning to collect data and quantify real-time changes in both LM (length) and CM contraction (diameter) amplitudes. Multiple images with 24 image enhancement filters and lighting from various directions are taken and analyzed to detect patterns, such as blood vessels or mesentery, on the intestinal surface to optimize real-time visualization and tracking during intestinal movements, even when their orientation changes **Fig. 1C**). Build-in 'Auto-Teach Inspection Tools' allows the camera to 'learn' and identify variations and differences in the pattern that may occur with intestinal contraction in real-time, ensuring stable tracking to obtain quantifiable data acquisition of diameter and longitudinal movements. This capability enables real-time tracking with less noise during data capture.

3. Can the authors provide assurance that the incorporation of pressure sensors at both tissue extremities does not impede or alter the natural movement of the intestine?

Response: Thank you for the suggestion and the paragraph has been appropriately modified and included in the text. In the current study, the oral and aboral ends of the tissues were connected to two individual pressure ports at the top of the perfusion bath by using a Tygon tubing with an inner diameter of 0.79 mm size to a differential pressure transducer MPX (type 399/2) that uses a monolithic silicon piezoresistor (Hugo Sachs Elektronik/Harvard Apparatus, Germany). The pressure was calibrated after adjusting the water column using the afterload control (**Fig. 2**). This setup therefore accurately delivers pressure changes occurring in the intestinal segment (PBT0). Such a setup does not impede or alter the natural movement of the intestine.

4. It would be informative to know the viable duration for which the intestine remains functional post-dissection.

Response: Thank you for the suggestion. Most of the intestinal segments studied stayed viable for ~ 45 minutes. During this period, the ILP and volume changes were maintained grossly at a steady level. As the tissue deteriorated in the chamber there were gross reduction in ILP and increase in volume with time. Therefore, most of the studies were limited into the first 30 minutes. The above text has now been added to the Method section.

5. On Page 3, references 7-10 seem to overlook some of the recent advancements in intestinal imaging. I recommend considering the inclusion of the following studies:

Response: Thank you for the suggestion and the references have now been included.
Wang, D., et al. (2021). Trans-illumination intestine projection imaging of intestinal motility in mice. *Nature Communications*, 12(1), 1682.
Boquet-Pujadas, A., et al. (2022). 4D live imaging and computational modeling of a functional gut-on-a-chip evaluate how peristalsis facilitates enteric pathogen invasion. *Science Advances*, 8(42), eabo5767.

6. The manuscript references a "proprietary program, peristalsis.exe" but offers limited details about its innovative features. Could you elucidate how the "predetermined thresholds" were established? Additionally, it would be beneficial to know if this program has undergone validation against traditional methodologies.

Response: The graph obtained from the BDAS software for ILP data plots ILP (in cmH₂O) against time. First, we manually calculated the amplitude of waves from several experiments and found the pressure levels ranging from 0.03 to 0.15 cmH₂O were considered low amplitude pressure waves, while pressures exceeding 0.15 cmH₂O indicated high amplitude contractions. respectively. Any amplitude difference less than 0.03 is considered as a flat line (no wave region). The frequencies obtained from these low and high amplitude waves matched with that of similar studies by other authors. It was also observed that the high amplitude waves occurred at irregular intervals and had similar duration as that of the low amplitude waves.

REVIEWERS' COMMENTS:

Reviewer #2 (Remarks to the Author):

The authors have addressed all my concerns. The addition of a flow chart is particularly helpful. The manuscript can now be accepted.